

# Distance measurement between trityl radicals by pulse dressed electron paramagnetic resonance with phase modulation

Nino Wili[1], Henrik Hintz[2], Agathe Vanas[1], Adelheid Godt[2], and Gunnar Jeschke[1]

[1]Department of Chemistry and Applied Biosciences, Laboratory of Physical Chemistry, ETH Zurich, Vladimir-Prelog-Weg 2, 8093 Zurich, Switzerland
[2]Faculty of Chemistry and Center for Molecular Materials (CM2), Bielefeld University, Universitätsstrasse 25, 33615 Bielefeld, Germany

**Correspondence:** nino.wili@phys.chem.ethz.ch, https://orcid.org/0000-0003-4890-3842

**Abstract.**

Distance measurement in the nanometer range is among the most important applications of pulse electron paramagnetic resonance today, especially in biological applications. The longest distance that can be measured by all presently used pulse sequences is determined by the phase memory time $T_m$ of the observed spins. Here we show that one can measure the dipolar coupling *during* strong microwave irradiation by using an appropriate frequency- or phase-modulation scheme, *i.e.* by applying pulse sequences in the nutating frame. This decouples the electron spins from the surrounding nuclear spins and thus leads to significantly longer relaxation times of the microwave-*dressed* spins (*i.e.* the rotating frame relaxation times $T_{1\rho}$ and $T_{2\rho}$) compared to $T_m$. The electron-electron dipolar coupling is not decoupled as long as both spins are excited, which can be implemented for trityl radicals at Q-band frequencies (35 GHz, 1.2 T). We show results for two bis-trityl rulers with inter-electron distances of about 4.1 nm and 5.3 nm and discuss technical challenges and possible next steps.

## 1 Introduction

Pulsed dipolar electron paramagnetic resonance (EPR) spectroscopy emerged as a powerful tool to measure distance distributions between electron spins in the nanometer range (Jeschke, 2012). This information is particularly useful when studying molecules and molecule assemblies that are intrinsically disordered or partially disordered or otherwise hard to crystallize and difficult to study with NMR or cryo-EM alone, *e.g.* certain membrane proteins (Bordignon and Bleicken, 2018) or protein-RNA complexes (Duss et al., 2014). The distance information is encoded in the magnetic dipole-dipole coupling between the electron spins, which depends on the inverse cubed distance, $r^{-3}$. A plethora of different techniques have been introduced, most notably double electron electron resonance (DEER) (Milov et al., 1984; Pannier et al., 2000), double quantum coherence (DQC) (Borbat and Freed, 1999), the single frequency for refocusing (SIFTER) (Jeschke et al., 2000), and relaxation induced dipolar modulation enhancement (RIDME) (Kulik et al., 2001; Milikisyants et al., 2009). The limiting factor for all these pulse sequences is the electron phase memory time $T_m$, which determines the maximal dipolar evolution time and thus the longest distance that can be measured. In many cases, the phase memory time can be prolonged by deuterating the solvent, or even the



whole protein (Ward et al., 2010; Schmidt et al., 2016). However, such an approach is costly and is rarely feasible, *e.g.* it is very difficult for membrane proteins in a lipid bilayer and impossible or in-cell work.

In recent years, several groups tried to use dynamical decoupling sequences based on multiple refocusing pulses (also known as Carr-Purcell sequences) in order to prolong the coherence times (Borbat et al., 2013; Spindler et al., 2015). Although shaped pulses significantly improved the fidelity of EPR experiments, pulse frequency band overlap and non-uniform inversion are still a problem in these sequences and can lead to artifacts, which may be corrected if traces with sufficient signal-to-noise and only moderately decaying background can be acquired (Breitgoff et al., 2017). Nevertheless, the improvements in $T_m$ so far are on the order of a factor of 2, which only marginally (though sometimes decisively) improves the longest attainable distance.

Recently, a sequence based on spin-diffusion, which would be limited by $T_1$ rather than $T_m$, was proposed (Blank, 2017). This experiment is still waiting for experimental verification.

Here we propose a sequence where the longest dipolar evolution time is, in principle, limited by the rotating frame relaxation time $T_{2\rho}$, which is often much longer than $T_m$ (for a discussion of $T_{2\rho}$ vs. the more familiar $T_{1\rho}$, *vide infra*). The complete dipolar evolution takes place *during* strong microwave irradiation. This decouples the electron spins from the surrounding nuclei (Jeschke and Schweiger, 1997) while the electron-electron coupling is still active. The spin manipulation during the strong microwave irradiation is achieved by short intervals of sinusoidal phase modulation. The frequency of this modulation needs to match the Rabi or nutation frequency of the spin-locking irradiation.

The latter approach was discovered more than once in the history of magnetic resonance. It traces back to investigations of Redfield on "rotary saturation" (Redfield, 1955). Hoult introduced the related idea of longitudinal field modulation for nutation frequency selective pulses to MRI (Hoult, 1979). Grzesiek and Bax picked up Hoult's idea, but used a phase modulation scheme instead and applied it to homonuclear mixing in solution state NMR (Grzesiek and Bax, 1995). They termed the technique "Audio-frequency NMR in a nutating frame", because their phase modulation (PM) frequency is in the audible range, and the pulse sequences effectively take place in a frame that nutates with the Rabi frequency of the spin-lock. Independently, Jeschke used longitudinal field modulation during a spin-lock for pulse EPR (Jeschke, 1999) and used the term "dressed EPR", because the spins are dressed by the microwave field during the spin-lock. This term is borrowed from quantum optics (Cohen-Tannoudji et al., 1992). The idea of dressed EPR originated in artifacts in hyperfine-decoupled electron-nuclear double resonance (ENDOR) spectra, which appear if the radio-frequency coil is not aligned perfectly perpendicular to the static field (Jeschke and Schweiger, 1997). Much later, it was also realized that field modulation should also prolong Rabi oscillations in the presence of inhomogeneous microwave fields (Saiko et al., 2018). Recently, the quantum information processing community picked up the idea of dressing electron spins in order to prolong coherence times (Laucht et al., 2016, 2017; Cohen et al., 2017). During the writing of this manuscript, Chen and Tycko came up with the idea of phase-modulation during a spin-lock independently again, and used it for slice selection during off-resonance spin-locks in solid-state, DNP-enhanced MRI (Chen and Tycko, 2020).

Here we combine the ideas of applying pulse sequences on dressed spins (Grzesiek and Bax, 1995; Jeschke, 1999) with the one of prolonging coherence times as a means of improving distance distribution resolution or prolonging distance range in pulsed dipolar EPR spectroscopy. To test the method, we used two bis-trityl rulers in which two trityl radicals are connected by



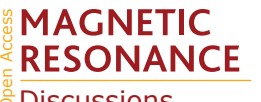

a rather stiff linker. Linker length and residual flexibility are known (Godt et al., 2006; Jeschke et al., 2010). The chosen trityl radical is structurally closely related to the Finland trityl radical and has similar EPR spectroscopic properties (Hintz et al., 2019). The narrow EPR spectrum of the used trityl radical makes it particularly amenable to single-frequency techniques for measurements of the dipole-dipole coupling (Reginsson et al., 2012) in a regime that is analogous to the one of homonuclear NMR experiments.

The article is organized as follows: First, we review mathematically, in the language of the magnetic resonance community, what happens to all the interactions in the spin Hamiltonian if we apply a strong microwave field. In order to do this, we will introduce a nutating frame description. Then we explain how an appropriate phase modulation scheme leads to "pulses" in the nutating frame. In the results section we show the synthesis of the bis-trityl rulers and present the application of a dressed spin echo experiment to such rulers to measure the dipolar coupling between two trityl radicals.

## 2 Theory

We use the following convention for operators: No prime refers to the laboratory frame and one prime to the electron-spin rotating frame, *i.e.* the interaction frame with the Zeeman Hamiltonian of the electrons. Two primes refer to the nutating frame, which is obtained with an additional interaction frame transformation with the pulse Hamiltonian. We will usually only denote the Hamiltonian with primes, and not all operators. If we mention axes in the text, we will explicitly use the primes, but we will omit them in mathematical formulas.

### 2.1 Averaging of interactions by strong continuous microwave irradiation

In order to understand the observations in this work, we need to study the influence of strong microwave irradiation on the different interactions present in the spin system. The spin Hamiltonian of a system with two coupled electrons ($S = 1/2$) in a bath of nuclei is given in the rotating frame by

$$\hat{\mathcal{H}}' = \hat{\mathcal{H}}'_{\mathrm{mw}} + \hat{\mathcal{H}}'_{\mathrm{offset}} + \hat{\mathcal{H}}'_{\mathrm{e\text{-}e}} + \hat{\mathcal{H}}'_{\mathrm{e\text{-}n}} + \hat{\mathcal{H}}'_{\mathrm{nuc}} \quad . \tag{1}$$

The first term is the microwave Hamiltonian, which is given in the electron-spin rotating frame by

$$\hat{\mathcal{H}}'_{\mathrm{mw}} = \omega_1 \left( \hat{S}_{1,x} + \hat{S}_{2,x} \right) \text{ with } \omega_1 = -\gamma_e B_1 \quad . \tag{2}$$

The Rabi or nutation frequency is denoted by $\omega_1$, which depends on the microwave amplitude $B_1$ and the gyromagnetic ratio of the electron, $\gamma_e$. We assume a constant microwave phase and neglect the influence of the microwaves on the nuclear spins. In the following, we will apply an interaction frame transformation (IAT) with $\hat{\mathcal{H}}'_{\mathrm{mw}}$ to all other terms and use first-order average Hamiltonian theory to gain physical insight. The new frame is referred to as the nutating frame. The nutating frame Hamiltonian is based on spin operators for dressed electron spins and bare nuclear spins. For mathematical details please consult the SI.

If we choose the nutating frame frequency $\omega_{\mathrm{PM}}$ equal to the Rabi frequency, $\omega_{\mathrm{PM}} = \omega_1$, the irradiation term is completely absorbed into the frame. In a real experiment with an ensemble of spins, $\omega_1$ will be distributed due to microwave inhomogeneities,



thus we will always have a remaining contribution of

$$\hat{\mathcal{H}}''_{\mathrm{mw}} = \Omega_{\mathrm{d}} \left( \hat{S}_{1,x} + \hat{S}_{2,x} \right) \text{ with } \Omega_{\mathrm{d}} = (\omega_1 - \omega_{\mathrm{PM}}) \quad . \tag{3}$$

The dressed spin offset $\Omega_{\mathrm{d}}$ will be distributed over the sample, but as a molecule is by orders of magnitude smaller than the microwave wavelength, $\Omega_{\mathrm{d}}$ will be the same for all electron spins within one molecule.

As usual, the influence of a small $g$-anisotropy and of an inhomogeneous static magnetic field $B_0$ is captured in offset terms in the rotating frame

$$\hat{\mathcal{H}}'_{\mathrm{offset}} = \Omega_{S,1} \hat{S}_{1z} + \Omega_{S,2} \hat{S}_{2z} \quad . \tag{4}$$

We neglect any tilt of the electron spin quantization axis due to strong $g$-anisotropy, which is a good approximation for trityl and other organic radicals. The first-order average Hamiltonian after an IAT with $\hat{\mathcal{H}}'_{\mathrm{mw}}$ vanishes, *i.e.*

$$\hat{\mathcal{H}}''_{\mathrm{offset}} = 0 \quad . \tag{5}$$

In pulse EPR, the spectral width is often much larger than the Rabi frequency. In this case, the first order approximation will be poor. It is, however, not a poor approximation for trityl radicals with our setup. For simplicity, we will mostly neglect the
effect of resonance offsets $\Omega_{S,1}$ and $\Omega_{S,2}$.

The most important term in the context of distance measurements is the electron-electron coupling Hamiltonian, which contains dipolar and exchange ($J$) contributions

$$\hat{\mathcal{H}}'_{\mathrm{e\text{-}e}} = \hat{\mathcal{H}}'_{\mathrm{e\text{-}e,dip}} + \hat{\mathcal{H}}'_{\mathrm{e\text{-}e,J}}$$
$$\hat{\mathcal{H}}'_{\mathrm{e\text{-}e,dip}} = \omega_{\mathrm{dd}} \left( \hat{S}_{1z} \hat{S}_{2z} - \frac{1}{2} \left( \hat{S}_{1x} \hat{S}_{2x} + \hat{S}_{1y} \hat{S}_{2y} \right) \right)$$
$$\omega_{\mathrm{dd}} = \frac{\mu_0}{4\pi} \frac{\mu_{\mathrm{B}}^2 g_1 g_2}{\hbar} \frac{1}{r_{12}^3} \left( 1 - 3\cos^2 \theta \right)$$
$$\hat{\mathcal{H}}'_{\mathrm{e\text{-}e,J}} = J \left( \hat{\boldsymbol{S}}_1 \cdot \hat{\boldsymbol{S}}_2 \right) \tag{6}$$

where $\mu_0$ is the vacuum permeability, $\mu_{\mathrm{B}}$ the Bohr magneton, $g_1$ and $g_2$ are the $g$-factors of the two electron spins, and $\theta$ is the angle between the external magnetic field and the interspin vector with length $r_{12}$. The exchange contribution is often, but not always negligible in pulse EPR based distance measurements. The prefactor of the dipolar coupling contains the distance
information and is given by

$$d = \frac{1}{2\pi} \frac{\mu_0}{4\pi} \frac{\mu_{\mathrm{B}}^2 g^2}{\hbar} \frac{1}{r^3} \quad . \tag{7}$$

This amounts to 52.04 MHz for $r = 1$ nm. After transformation to the nutating frame, we obtain

$$\hat{\mathcal{H}}''_{\mathrm{e\text{-}e,dip}} = -\frac{1}{2} \cdot \omega_{\mathrm{dd}} \left( \hat{S}_{1x} \hat{S}_{2x} - \frac{1}{2} \left( \hat{S}_{1z} \hat{S}_{2z} + \hat{S}_{1y} \hat{S}_{2y} \right) \right)$$
$$\hat{\mathcal{H}}''_{\mathrm{e\text{-}e,J}} = \hat{\mathcal{H}}'_{\mathrm{e\text{-}e,J}} = J \left( \hat{\boldsymbol{S}}_1 \cdot \hat{\boldsymbol{S}}_2 \right) \quad . \tag{8}$$





The electron-electron dipolar coupling is not averaged to zero, but only scaled by a factor of -1/2. It is also tilted such that the unique axis of the coupling Hamiltonian points along the spin-lock axis ($z' \rightarrow x'' = x'$, in the NMR literature, often a tilted frame is used). In other words, the two dressed spins are still dipole-dipole coupled with half the original coupling strength and with inverted sign of the interaction. This result is well-known in solid-state NMR (Rhim et al., 1970), where it is used to generate "magic echoes". The isotropic $J$-coupling is unaffected if both spins are irradiated. Note however that the difference

of the resonance frequencies of the two dressed spins is much smaller than the one of the bare spins, as remarked upon already by Grzesiek and Bax (Grzesiek and Bax, 1995). The difference in relative magnitude of the exchange coupling and resonance frequency difference can lead to a different manifestation of the exchange coupling in the spectra.

The term $\hat{\mathcal{H}}'_{\text{e-n}}$ contains all electron-nucleus (hyperfine) couplings. If the Rabi frequency of the irradiation is much larger than all hyperfine couplings and nuclear Zeeman frequencies, this term also averages to zero in the nutating frame, *i.e.*

$$\hat{\mathcal{H}}''_{\text{e-n}} = 0 \quad , \tag{9}$$

an effect referred to as hyperfine decoupling (Jeschke and Schweiger, 1997). Terms that do not contain an electron spin operator are assumed to be unaffected by the microwave irradiation,

$$\hat{\mathcal{H}}_{\text{nuc}} = \hat{\mathcal{H}}'_{\text{nuc}} = \hat{\mathcal{H}}''_{\text{nuc}} \quad . \tag{10}$$

Equations (9) and (10) might appear to be irrelevant to distance measurements between electrons, but they are not. The terms

$\hat{\mathcal{H}}_{\text{e-n}}$ and $\hat{\mathcal{H}}_{\text{nuc}}$ do not commute if nuclear-nuclear flip-flop terms are present, even if the hyperfine coupling $\hat{\mathcal{H}}_{\text{e-n}}$ is purely secular (no electron spin echo envelope modulation effect). For example, for the flip-flop terms in $\hat{\mathcal{H}}_{\text{nuc}}$, $\left[ \hat{S}_z \hat{I}_{iz}, \hat{I}_i^+ \hat{I}_j^- \right] \neq 0$. A simple spin echo sequence on the electron spins thus does *not* completely refocus the hyperfine coupling - the result is a dephasing of the electron spins, sometimes loosely referred to as "relaxation". In principle, this dephasing stems from coherent evolution, but since the nuclear spin bath is usually very large, it is computationally very expensive to simulate a real system.

Accordingly, most theoretical studies treat the internuclear couplings phenomenologically using effective flip rates (Klauder and Anderson, 1962). The situation during microwave irradiation of an electron-nuclear spin system has many parallels with heteronuclear decoupling in solid-state NMR (Ernst, 2003), where one distinguishes between the "real" transverse relaxation time due to incoherent dynamics, $T_2$, and the effective relaxation time that is measured with a spin echo, $T_2'$ and has large coherent contributions. Of course, in EPR, the coupling strengths and Rabi frequencies are several orders of magnitude higher

than in NMR.

In EPR measurements of organic radicals at sufficiently low temperatures, usually at 50 K and below, the hyperfine and nuclear-nuclear couplings dominate the dephasing (Brown, 1979). In this case, averaging the hyperfine coupling to zero should drastically increase the dephasing time, because $\hat{\mathcal{H}}_{\text{nuc}}$ commutes with all remaining terms containing electron spin operators. At the same time, according to Eq. (8), the effective dipolar coupling is scaled by a factor -1/2. If the gain in dephasing time

is larger than a factor of two, it should - in principle - be possible to measure longer dipolar dephasing traces and thus longer distances. As we shall see later, the effective scaling factor may be even more favourable (-3/4), as the flip-flop terms in the electron-electron dipolar Hamiltonian may be truncated for bare spins but can be significant for dressed spins.



The immediate next question is then how one can measure the dipolar coupling during a spin-lock pulse. We propose to use a phase-modulation scheme that we discuss in the next section.

## 2.2 Pulse dressed spin resonance with phase-modulated pulses

The basic theory of dressed spin resonance is already described in (Grzesiek and Bax, 1995) and (Jeschke, 1999) but we describe it here again for completeness and consistency.

For simplicity and illustration, we first look at an isolated electron (spin 1/2) in a static magnetic field $B_0$ along the laboratory-frame $z$-axis. If we irradiate this system with a linearly polarized electromagnetic field with frequency $\omega_{\mathrm{mw}}$ and

amplitude $2B_1$, the Hamiltonian in angular frequency units is given by

$$\hat{\mathcal{H}} = \omega_0 \hat{S}_z + 2\omega_1 \cos\left(\omega_{\mathrm{mw}} t + \phi_{\mathrm{mw}}(t)\right) \hat{S}_x, \tag{11}$$

with $\omega_0 = -\gamma B_0$. We include an arbitrary phase $\phi_{\mathrm{mw}}(t)$, which we will use later to generate dressed spin PM pulse sequences. As usual, we now go into a rotating frame with frequency $\omega_{\mathrm{mw}}$. If we neglect the time-dependent terms (rotating wave approximation, RWA), we obtain

$$\hat{\mathcal{H}}' = \Omega_S \hat{S}_z + \omega_1 \left(\cos\left(\phi_{\mathrm{mw}}(t)\right) \hat{S}_x + \sin\left(\phi_{\mathrm{mw}}(t)\right) \hat{S}_y\right), \tag{12}$$

with the offset $\Omega_S = (\omega_0 - \omega_{\mathrm{mw}})$, which is also used in Eq. (4). The main effect of the time-dependent terms is a Bloch-Siegert shift, *i.e.* just a small correction of $\Omega_S$. We can choose the PM as

$$\phi_{\mathrm{mw}}(t) = \phi_0 + a_{\mathrm{PM}} \cos\left(\omega_{\mathrm{PM}} t + \phi_{\mathrm{PM}}\right), \tag{13}$$

with a modulation amplitude $a_{\mathrm{PM}}$, a modulation frequency $\omega_{\mathrm{PM}}$ and a modulation phase $\phi_{\mathrm{PM}}$. The phase $\phi_0$ is what one would

conventionally call the phase of the microwave pulse applied to the bare spins, *i.e.* $[0, \pi/2, \pi, 3\pi/2]$ for $[x, y, -x, -y]$. Likewise, $\phi_{\mathrm{PM}}$ is the phase of the PM pulse that is applied to the dressed spins. We use $\phi_0 = 0$ for the following discussion. For small modulation amplitudes, $a_{\mathrm{PM}} \ll 1$, we can use the approximations $\cos\left(\phi_{\mathrm{mw}}\right) \approx 1$ and $\sin\left(\phi_{\mathrm{mw}}\right) \approx \phi_{\mathrm{mw}}$ and obtain a truncated rotating-frame Hamiltonian

$$\hat{\mathcal{H}}' \approx \Omega_S \hat{S}_z + \omega_1 \hat{S}_x + \omega_1 a_{\mathrm{PM}} \cos\left(\omega_{\mathrm{PM}} t + \phi_{\mathrm{PM}}\right) \hat{S}_y. \tag{14}$$

For a hard pulse, *i.e.* $\omega_1 \gg \Omega_S$, we can now apply a second interaction frame transformation with $\omega_{\mathrm{PM}} \hat{S}_{x'}$, use the RWA again, and obtain the dressed rotating frame Hamiltonian

$$\hat{\mathcal{H}}'' = \Omega_{\mathrm{d}} \hat{S}_x + \frac{\omega_1 a_{\mathrm{PM}}}{2} \left(\cos\left(\phi_{\mathrm{PM}}\right) \hat{S}_y + \sin\left(\phi_{\mathrm{PM}}\right) \hat{S}_z\right), \tag{15}$$

with the dressed spin offset $\Omega_{\mathrm{d}} = (\omega_1 - \omega_{\mathrm{PM}})$, already introduced in Eq. (3), and a dressed spin nutation (Rabi) frequency of $\omega_1 a_{\mathrm{PM}}/2$. Again, the RWA implies that we neglect a Bloch-Siegert shift, now for the dressed spins, which would introduce a

correction to $\Omega_{\mathrm{d}}$. The whole situation is analogous to the rotating frame Hamiltonian in Eq. 12, but with an exchange of axes.





Some words of caution: First, in EPR unlike in NMR, the hard pulse limit will often not be fulfilled. In a first step, one can use an interaction frame transformation with the whole effective nutation field, $\Omega_S \hat{S}_z + \omega_1 \hat{S}_x$. For the sake of intuitive clarity, we will not do this for the qualitative discussion. Second, one can easily choose a large $a_{\mathrm{PM}}$, such that the RWA leading from Eq. (14) to Eq. (15) is seriously invalid. This was recognized already in (Grzesiek and Bax, 1995), and studied separately in

(Laucht et al., 2016). In our study, imperfection of the RWA is visible in nutation curves, but the final results do not seem to be affected. The problem might be alleviated by using an appropriate frequency or amplitude modulation in order to generate a circularly polarized field in the rotating frame.

There are two alternatives to the phase-modulation schemes. One could equivalently formulate the dressed spin resonance as a frequency-modulation. Phase- and frequency modulation are physically equivalent, but we prefer the phase-modulation be-

cause the description of frequency-modulation involves a time-dependent offset/detuning and thus a "wobbling" frame, which makes it harder to keep track of relative phases of coherences. Instead of any microwave/radio-frequency modulation, one could also use a modulation of the magnetic field along the laboratory frame $z$ direction (Jeschke, 1999). Depending on the setup, the relative phase of the modulation can be locked to the phase of the driving field or not. If an arbitrary waveform generator setup is available, phase modulation may be preferable, as it does not require modulation coils and a radiofrequency amplifier and

makes synchronization of bare-spin and dressed-spin pulses much easier. However, the amplitude of the phase pulses depend on the Rabi frequency itself in the case of phase modulation. By using an external oscillating field, this dependence would vanish.

## 2.3 Pulse sequence

The pulse sequence used to measure the dipolar coupling in this work is the dressed-spin primary echo sequence shown in Fig.

1. It can be readily understood with results from the previous sections. For dipolar measurements, one chooses $\tau_1 = \tau_2$ and constant $T_{\mathrm{SL}}$. The first $\pi/2$ pulse generates electron coherence. Since we deal with trityl radicals, the excitation can be nearly uniform on our setup. The spins are then locked with a spin-lock pulse that is 90 degrees phase shifted with respect to the first pulse. Let us assume that the spin-lock and the coherences are along $x'$. For free dressed-spin evolution, *i.e.* in the absence of phase modulation, we can assume the following Hamiltonian during the spin-lock in the nutating frame:

$$\hat{\mathcal{H}}'' = \Omega_{\mathrm{d}} \left( \hat{S}_{1x} + \hat{S}_{2x} \right)$$
$$- \frac{\omega_{\mathrm{dd}}}{2} \left( \hat{S}_{1x} \hat{S}_{2x} - \frac{1}{2} \left( \hat{S}_{1z} \hat{S}_{2z} + \hat{S}_{1y} \hat{S}_{2y} \right) \right), \tag{16}$$

where we recall that $\Omega_{\mathrm{d}} = (\omega_1 - \omega_{\mathrm{PM}})$. Note that $\omega_1$ is inhomogeneous over the sample, but is the same within each pair of spins.

The Hamiltonian in Eq. 16 is analogous to the one in the rotating frame, but to a very good approximation the offsets are

the same for both dressed electron spins. Additionally, all hyperfine couplings vanish. The phase modulation pulses act on the dressed spins in the nutating frame. We can thus generate a *dressed spin echo* by a phase-pulse sequence $\pi/2 - t - \pi - t$. A third $\pi/2$ pulse is needed that rotates any refocused dressed-spin coherence back to the $x'' = x'$ axis. The magnetization resulting





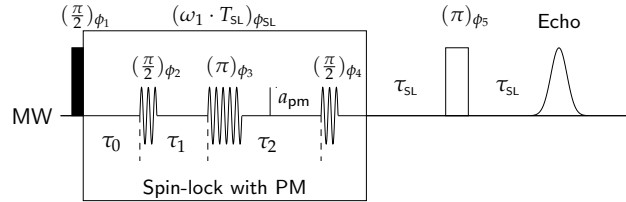

**Figure 1.** Pulse sequence used to measure the dipolar coupling during a spin-lock. Note that $|\phi_{\text{SL}} - \phi_1| = \pi/2$. The phases $\phi_{2-4}$ correspond to $\phi_{\text{PM}}$ in Eq. 13, while $\phi_{\text{SL}} = \phi_0$ in the same equation. Details for the inner working of the sequence are given in the main text.

from this backrotation is locked again, until it is detected by the remaining $\tau - \pi - \tau-$ echo sequence. A very similar sequence was already demonstrated with $z$-modulation pulses in (Jeschke, 1999), albeit not for dipole-dipole coupled electron spins.

The dressed-spin echo is needed to refocus microwave field inhomogeneities (*i.e.* a distribution of $\omega_1$ and thus also $\Omega_{\text{d}}$). The dipolar part of the Hamiltonian is unaffected by the PM-$\pi$ pulse, because this pulse inverts both spin operators at the same time. With effects of the other terms being refocused, it is sufficient to only keep the dipolar part during the periods $\tau_1$ and $\tau_2$:

$$\tilde{\tilde{\mathcal{H}}}'' = -\frac{\omega_{\text{dd}}}{2} \left( \hat{S}_{1x}\hat{S}_{2x} - \frac{1}{2} \left( \hat{S}_{1z}\hat{S}_{2z} + \hat{S}_{1y}\hat{S}_{2y} \right) \right). \tag{17}$$

At the start of the period $\tau_1$, the system is in the state $\hat{\sigma}'' = \hat{S}_{1z} + \hat{S}_{1z}$ (or along $y''$, depending on the phase $\phi_2$). For $\tau_1 = \tau_2$,

this evolves according to

$$
\hat{\sigma}'' \xrightarrow{\tilde{\tilde{\mathcal{H}}}'' \cdot 2\tau_1} \cos\left(\frac{3}{4}\omega_{\text{dd}}\tau_1\right) \left( \hat{S}_{1z} + \hat{S}_{1z} \right)
$$
$$
+ \sin\left(\frac{3}{4}\omega_{\text{dd}}\tau_1\right) \left( 2\hat{S}_{1x}\hat{S}_{1y} + 2\hat{S}_{1y}\hat{S}_{1x} \right) \quad . \tag{18}
$$

The $z''$ terms are then flipped to $x'' = x'$, are transferred to bare-spin coherence at the end of the microwave pulse and are then detected by the echo. The other terms do not contribute to the detected signal. The factor of $3/4$ has two contributions. A

factor of $(-)1/2$ is due to the spin-lock and the partial averaging of the dipolar coupling, see Eq. (8). A factor of $3/2$ is due to the strong coupling regime in the dressed frame, because the dressed electron spins are equivalent. This scaling by a factor of $3/2$ for trityl biradicals has been observed before at short distances with established single-frequency techniques (Meyer et al., 2018), where it results from the dipole-dipole coupling being much larger than the mean difference of the resonance frequencies of two trityl radicals. In conclusion, we expect that for dressed spins echo intensity oscillates with $3/4$ of the

dipolar coupling, which for a fixed or narrowly distributed distance will manifest in a Pake pattern because the measurements are conducted in frozen solution.

The timing $\tau_{\text{SL}}$ of the read-out echo does, in principle, affect the resulting dipolar spectrum, because it acts as filter with the signal intensity scaling with $\cos\omega_{\text{dd}}\tau_{\text{SL}}$. However, for short interpulse delays and long distances, such filtering should be negligible. If necessary, a SIFTER-type readout sequence could be used, which refocuses both the offsets and the dipolar couplings.

couplings.



It is noteworthy that, in principle, a normal two-pulse echo on the bare spins with non-selective pulses would be sufficient to measure the dipolar coupling. In practice, this approach is usually much inferior to the DQC and SIFTER sequences, because the phase memory time is of the same order of magnitude as the dipolar oscillations, echo decay is not monoexponential and contains other contributions, and dead-time is significant. The combination of these complications makes it very difficult to separate the dipolar oscillation. Under the spin-lock, the relaxation is sufficiently slowed down, such that the dipolar evolution is clearly distinguishable, and the dead time in a PM-pulse sequence is nearly zero.

### 2.4 Expected limitations

The derivation of the modulation formula in Eq. (18) depends on the condition that $\omega_1$ is much bigger than all other frequencies present in the system. Especially for the bare-spin resonance offsets, this approximation is not fulfilled very well. In principle, one could account for the different offsets analytically, but this is rather tedious and does not provide much additional insight. We will present numerical simulations in the result section to illustrate the deviations.

## 3 Materials and methods

All measurements were performed on a home-built Q-band spectrometer equipped with a Keysight M8190A arbitrary waveform generator operating at 8 GS/s and an ADC with a sampling frequency of 2 GHz (SP Devices ADQ412) (Doll, 2016). The highly flexible software made it straightforward to implement the pulse sequences with PM pulses, in contrast to commercial analogues. Microwave pulses were amplified with a travelling wave tube (TWT) amplifier with 150 W nominal output power. A home-built Q-band loop-gap resonator for 1.6 mm tubes was used (Tschaggelar et al., 2017).

As model compounds, we used bis-trityl rulers with electron-electron distances of about 4.1 nm and 5.3 nm. The synthesis is discussed in section 4.1. The bis-trityl rulers were dissolved in *ortho*-terphenyl (OTP) or its perdeuterated analogue dOTP providing solutions of different concentrations. More details are given in each figure and the SI.

Measurements were conducted at 50 K using a liquid helium flow cryostat. We did not systematically test the optimal temperature for each measurement. However, it is likely that higher temperatures would allow for shorter shot repetition times without dramatically changing the dephasing times.

Frequency-domain spectra were measured with chirp echoes and subsequent Fourier transform instead of field sweeps (Doll and Jeschke, 2014). Chirp pulses covered a range of 300 MHz symmetrically around the centre of the spectrum. The powder spectrum was simulated with the *EasySpin* library (Stoll and Schweiger, 2006).

The two-pulse dephasing time $T_m$ was measured with a sequence $\pi/2 - \tau - \pi - \tau - $ echo with $t_\pi = 2t_\pi$. Different pulse lengths were used to check whether instantaneous diffusion contributes to coherence loss. Similar to previous findings by Meyer et al. (2018), it was found that flip angles of $\pi/2$ or $3\pi/2$ for the second pulse gave higher echo intensities than an angle of $\pi$. More details are given in the SI.



The rotating frame relaxation time $T_{1\rho}$ was measured with the sequence in Fig. 1 in the absence of any phase-modulation pulses and variable $T_{\text{SL}}$ and with $\tau_{\text{SL}} = 200$ ns. Interestingly, $T_{1\rho}$ is significantly different when measured with a simple spin-locked echo with the sequence $\pi/2 - \tau - \text{lock} - \tau - \text{echo}$. More details are given in the results section and the SI.

The rotating frame relaxation time $T_{2\rho}$ for the mono-trityl was measured with the sequence in Fig. 1 including the phase-modulation pulses and fixed $T_{\text{SL}}$ and with $\tau_{\text{SL}} = 200$ ns. In the case of the bis-trityls, it is impossible to measure $T_{2\rho}$ independent of the dipolar coupling. Where applicable, we mention the decay rate of the "intramolecular background" for comparison.

All decay rates were obtained by fitting a stretched exponential of the functional form

$$f(t) = \exp\left(-(t/T)^{\xi/3}\right) \tag{19}$$

to the relaxation curves, where $t = 2\tau$ and $T = T_m$ for the two-pulse echo decay, $t = 2\tau_1$ and $T = T_{2\rho}$ for dressed echo decays, and $t = T_{\text{SL}}$ and $T = T_{1\rho}$ for the longitudinal rotating frame relaxation time.

The Rabi frequency $\omega_1$ was measured with a nutation experiment $t_{\text{nut}} - T - \pi/2 - \tau - \pi - \tau - \text{echo}$. As a control, we performed a dressed-spin resonance experiment with the sequence in Fig. (4), but only one PM pulse with low amplitude and variable frequency. This also yields the $\omega_1$ spectrum (see the SI). A similar experiment with $z$-modulation was demonstrated in (Jeschke, 1999).

When the Rabi spectrum is known, one can set the value of the PM frequency $\omega_{\text{PM}}$. One then needs to choose a value for the modulation amplitude $a_{\text{PM}}$ and set up the PM pulse lengths. This can be achieved with a PM nutation experiment. Again, one uses the basic sequence in Fig. 1, with one pulse only with now fixed $\omega_{\text{PM}}$. One then observes the echo intensity as a function of the PM pulse length. That way the optimal PM pulse length can be determined. When choosing $a_{\text{PM}} = 0.3$, we observed only slight Bloch-Siegert shift related oscillations in the PM pulse nutation traces while achieving a PM $\pi$-pulse length of 40-42 ns.

The dressed echo can not be detected directly, because $\tau_1$ and $\tau_2$ are both indirect variables. Only the actual echo at the end of the microwave pulse sequence is digitized continuously. In order to optimize indirect detection, we checked that the last PM pulse in Fig. 1 is applied at the correct position. We observed that the position seems to be nearly perfectly predictable by setting $\tau_2 = \tau_1 + t_{\pi/2}$, where $t_{\pi/2}$ refers to the length of the PM-$\pi/2$-pulse. We observed crossing dressed-spin echoes when changing interpulse delays in the PM pulse sequence, similar to what is known in microwave multi-pulse sequences in pulse EPR. Interestingly, the position of some unwanted echoes depends on the choice of $\tau_0$. Nevertheless, all these unwanted echoes can be suppressed by phase cycling the initial phases $\phi_{\text{PM}}$ of the PM pulses, $\phi_{2-4}$.

A step-by-step guide to setting up the sequence is provided in the SI.

# 4 Results

## 4.1 Synthesis

The synthesis of bis-trityl rulers **1** and **2** is presented in Fig. 2. They were assembled from the rodlike building blocks **6** equipped with amino groups at both ends and trityl acid chloride **8**. The latter was prepared from the corresponding trityl acid **7** (also named mono-trityl) using a procedure that has been described for the corresponding conversion of the structurally related





Finland trityl radical (Shevelev et al., 2014). To achieve a complete conversion of the building blocks **6**, trityl acid chloride **8**
was used in excess. Leftover trityl acid chloride **8** is hydrolysed upon workup and the resulting trityl acid **7** is easily removed by
filtration through silica gel. The building blocks **6** were obtained through a sequence of alkynyl-aryl coupling reactions (Sahoo
et al., 2010; Qi et al., 2016; Ritsch et al., 2019) and a final oxidative alkyne dimerization. Oxidative alkyne dimerization is
a very efficient way to obtain rod-like spacers with the same functional groups at both ends. Although this gives a butadiyne
moiety, the spacer is still rather stiff and therefore the spin-spin distance sufficiently well-defined (Godt et al., 2006; Jeschke
et al., 2010).

### 4.2    Relaxation of mono-trityl 7

As a reference, we measured the spectrum and the relaxation properties of the mono-trityl **7** in dOTP, see Fig. 3. As visible in
panel (b), $T_{1\rho}$ is orders of magnitude larger than $T_m$. Unfortunately, our TWT prevents us from using spin-lock pulses of more
than 40 µs, meaning that uncertainty in $T_{1\rho}$ is rather large. Nevertheless, fitting a single stretched exponential to each curve
yields values of $T_m =$2.9 µs and $T_{1\rho} \approx$930 µs. As mentioned above, the distance measurements based on dressed spin echoes
are limited by the transverse rotating frame relaxation time $T_{2\rho}$ rather than the longitudinal one $T_{1\rho}$. The blue curve in panel
(b) shows the dressed echo decay, indicating that $T_m < T_{2\rho} \ll T_{1\rho}$, with a fitted value of $T_{2\rho} =$13.1 µs.

While conceptually simple, the large difference between $T_{1\rho}$ and $T_{2\rho}$ was rather surprising to us. We are not aware of any
example in the literature where $T_{2\rho}$ is discussed in-depth in the context of EPR, although there are several discussion in NMR
and MRI (Michaeli et al., 2004). It remains unclear what the limiting contribution to $T_{2\rho}$ is. In analogy to solid-state NMR,
residual coupling terms of the hyperfine interactions certainly contribute. An additional contribution would be the remaining
intermolecular dipolar couplings, but then we would expect a strong dependence on the concentration, which we did not
observe. Another factor that will definitely contribute is the noise of the driving field. The noise (phase and amplitude) of the
TWT during spin-lock will not be refocused by the dressed echo. It is hard to quantify this contribution, since we do not have
high-power amplifiers with different noise figures. In the future, we might investigate the influence of artificially added driving
noise on $T_{2\rho}$.

The large difference between $T_{1\rho}$ and $T_{2\rho}$ is unfortunate, because our proposed sequence will be limited by the latter.
Nevertheless, one might come up with a sequence that will be limited by the former, longer relaxation time, and thus we
measured $T_{1\rho}$ also for the bis-trityl rulers.

### 4.3    Bis-trityl 1, $r \approx$ 4.1 nm

The results for bis-trityl **1** are shown in Fig. 4. The chirp echo FT-EPR spectrum is shown in panel (a). The spectrum consists
of a slightly asymmetric line with an FWHM of 16 MHz. The theoretical excitation profile of a 4 ns and an 8 ns microwave
pulse are overlaid, showing that the whole spectrum can be excited almost uniformly with rectangular pulses.

The relaxation measurements for $T_m$ and $T_{1\rho}$ are displayed in panel (b), and they show the same trends as in the case of
the mono-trityl. Note that the $T_m$ measurement displayed was done with 100/200 ns pulses. Otherwise, the dipolar oscillations
are already strongly visible in the two-pulse echo decay. It is immediately clear that the rotating frame relaxation time $T_{1\rho}$ is



**Figure 2.** Synthesis of the bis-trityl rulers **1** and **2**. For $n = 1$: (a) PdCl$_2$(PPh$_3$)$_2$, CuI, piperidine, THF, rt, 25 h, 84%; (Ritsch et al., 2019) (b) K$_2$CO$_3$, MeOH, CH$_2$Cl$_2$, rt, 14.5 h, 96%; (c) PdCl$_2$(PPh$_3$)$_2$, CuI, piperidine, THF, air, rt, 16 h, 36%; (d) SOCl$_2$, CHCl$_3$, 50 °C, 90 min, not isolated; (e) $^i$Pr$_2$NEt, CHCl$_3$, rt, 17 h, 40%. For $n = 2$: (a) PdCl$_2$(PPh$_3$)$_2$, CuI, piperidine, THF, rt, 46 h, 86%; (b) K$_2$CO$_3$, MeOH, CH$_2$Cl$_2$, rt, 14.5 h, 96%; (c) PdCl$_2$(PPh$_3$)$_2$, CuI, piperidine, THF, air, rt, 15.5 h, 65%; (d) SOCl$_2$, CHCl$_3$, 50 °C, 90 min, not isolated; (e) $^i$Pr$_2$NEt, CHCl$_3$, rt, 19 h, 64%. For further details see the SI part B. THF = tetrahydrofuran, TIPS = triisopropylsilyl, TMS = trimethylsilyl, rt = room temperature.

much longer than the phase-memory time, $T_{1\rho} \gg T_{\mathrm{m}}$. The phase memory time is about 3.3 µs, while after 40 µs of spin-lock, the echo intensity is still more than 90 % of its maximal value. A naive fit with a stretched exponential yields $T_{1\rho} \approx 560$ µs.

The modulation of the dressed-spin echo is displayed in panel (c). Clear oscillations are visible in the primary data. Since we do not currently have a model for the background, we fitted a stretched exponential to the data. This background is very





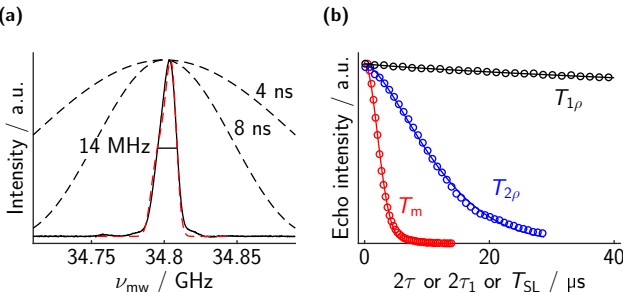

**Figure 3.** Measurements on mono-trityl **7**. (**a**) EPR spectrum. The excitation profiles of the rectangular pulses used are indicated. They are sufficiently strong to excite the whole EPR line. The red dashed lines indicate a simulation based on the $g$-values given in (Hintz et al., 2019) and an Gaussian broadening of 8 MHz FWHM. (**b**) Corresponding echo decay curves. Experimental points in circles (not all points shown for clarity), and best fit in solid lines. The fitted values are $T_m = 2.9\,\mu s$, $T_{2\rho} = 13.1\,\mu s$ and $T_{1\rho} = 930\,\mu s$.

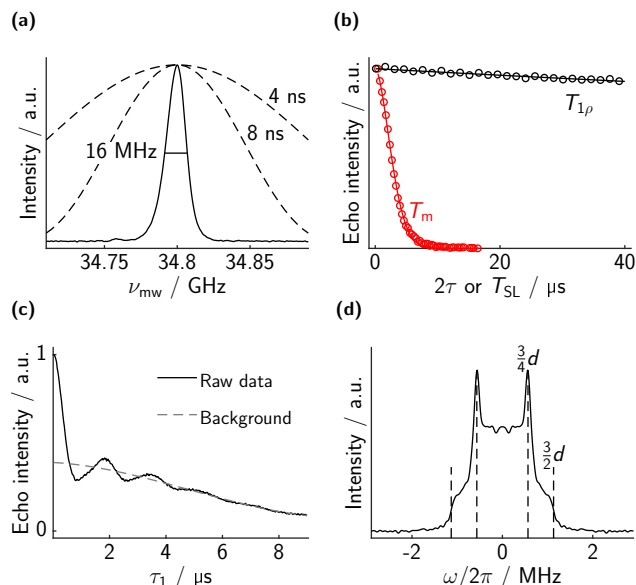

**Figure 4.** Measurements on bis-trityl **1**. (**a**) EPR spectrum. (**b**) Comparison of the decay of a microwave two-pulse echo (red, bare-spin decoherence) with the decay of the spin-locked echo as a function of $T_{SL}$ (black, dressed-spin polarization decay). Experimental points in circles (not all points shown for clarity), and best fit in solid lines. The fitted values are $T_m = 3.3\,\mu s$, and $T_{1\rho} = 560\,\mu s$ (**c**) Dressed-spin echo evolution as a function of $\tau_1 = \tau_2$. The dipolar oscillations are clearly visible. A stretched exponential background with $T_{2\rho} = 14.3\,\mu s$ is shown in gray. (**d**) Dipolar spectrum obtained by a Fourier transform of (**c**) after background division. The positions of the expected singularities based on the distance of the electrons are indicated by dashed lines. Note that there are small artifact peaks outside the plotting range at $\pm 8$ MHz which we suspect to be a sampling artifact.

similar to the $T_{2\rho}$ measurement of the mono-trityl (14.3 μs *vs.* 13.1 μs decay constant), which also means that it decays much



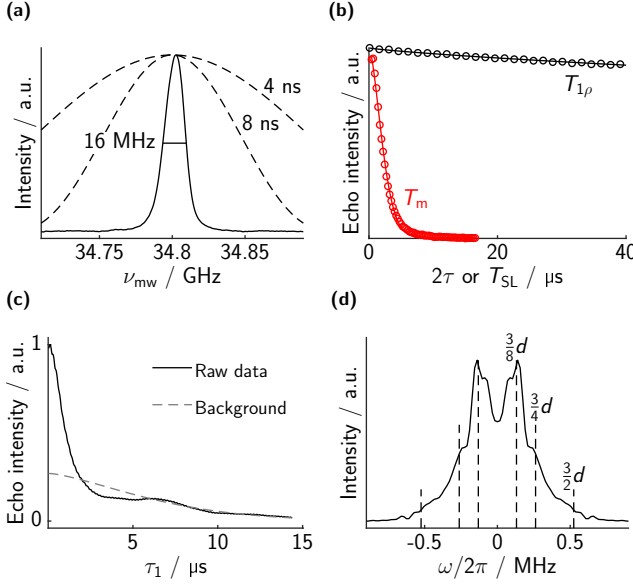

**Figure 5.** (Measurements on bis-trityl **2**. **a**) EPR spectrum. (**b**) Bare-spin decoherence (two-pulse echo decay, red) and dressed-spin polarization decay (spin-locked echo decay, black). Experimental points in circles (not all points shown for clarity), and best fit in solid lines. The fitted values are $T_m =2.6\,\mu s$, and $T_{1\rho} \approx 730\,\mu s$ (**c**) PM echo evolution as a function of $\tau_1 = \tau_2$. A stretched exponential background with $T_{2\rho} =14.5\,\mu s$ is shown in gray. The dipolar oscillations are damped and the background obscures the oscillations at long dipolar evolution times. (**d**) Dipolar spectrum obtained by a Fourier transform of (**c**) after background division. In addition to the singularities expected from our basic theoretical treatment strong singularities at $3/8d$ are apparent. These features are explained in the main text.

faster than $T_{1\rho}$. After background correction by division and a Fourier transform, we obtain the spectrum in Fig. 4 (d). The spectrum is a nice Pake pattern with the characteristic singularities at one and two times the dressed-spin dipolar frequency. The singularities appear at the expected positions. The splitting parameter $d$ can be calculated from the expected distance of 4.1 nm, but it is scaled by a factor of $3/4$ as discussed above.

### 4.4 Bis-trityl 2, $r \approx 5.3$ nm

The analogous data of bis-trityl **2** are displayed separately in Fig. 5. The chirp echo FT EPR spectrum looks essentially the same as for bis-trityl **1**, with the same slight asymmetry and an FWHM of 16 MHz.

The two-pulse microwave echo decay is slightly faster for bis-trityl **2** (2.9 μs vs. 3.3 μs). Again, it is difficult to really quantify a decoherence time that is not influenced by residual echo envelope contributions from intramolecular electron-electron coupling. Even with 100/200 ns pulses the excitation profile of the $\pi$-pulse is still larger than the dipolar coupling and some dipolar contribution to the echo envelope function is expected. The signal decay of dressed spin polarization under the spin-lock ($T_{1\rho} = 730\,\mu s$) is again much slower, and comparable to the case of bis-trityl **1**. Unfortunately, the dipolar oscillations in panel (c) are not as clear as in the case of shorter distances. Also, the background is already rather fast compared to the dipolar





frequencies (14.5 μs decay constant). In the dipolar spectrum, panel (d), it becomes clear that this case is more complicated,

because additional singularities appear at around $3/8 \cdot d$. These features must result from the breakdown of some approximation that we have made in our theoretical description. Most likely they are due to the finite strength of the spin-lock compared to the inhomogeneous spectral width. For two spins with different bare-spin resonance offsets, both the direction and magnitude of the effective field in the rotating frame differs. Accordingly, the two dressed spins have different resonance frequencies and quantization axes. Unless the dipole-dipole coupling is much larger than the frequency difference, it is significantly perturbed.

In order to give a more quantitative explanation, we will show simplified numerical simulations in the following.

### 4.5 Numerical Simulations

In order to understand the deviation of our experimental results from the theoretical expectation based on first-order average Hamiltonian theory (especially in the case of bis-trityl **2**), we performed simplified numerical simulations. In principle, one could simulate the complete sequence, including the time-dependent phase during the phase-pulses. We chose a simplified

route: We start with both spins along $z'$ and then calculate the expectation value of $\hat{S}_z = \hat{S}_{1z} + \hat{S}_{2z}$ during the spin lock using the Hamiltonian

$$
\begin{aligned}
\hat{\mathcal{H}}' =& \Omega_1 \hat{S}_{1z} + \Omega_2 \hat{S}_{2z} \\
& + \omega_{\mathrm{dd}}(r,\theta) \left( \hat{S}_{1z} \hat{S}_{2z} - \frac{1}{2} \left( \hat{S}_{1x} \hat{S}_{2x} + \hat{S}_{1y} \hat{S}_{2y} \right) \right) \\
& + 2\pi \cdot \nu_1(t) \left( \hat{S}_{1x} + \hat{S}_{2x} \right)
\end{aligned}
\tag{20}
$$

In order to refocus the nutation of the spins around the effective field, we invert the phase of the irradiation in the middle of the spin-lock, such that

$$
\nu_1(t) =
\begin{cases}
\nu_1(0), & \text{for } 0 \leq t < \tau_1 \\
-\nu_1(0), & \text{for } \tau_1 \leq t < 2\tau_1 \quad .
\end{cases}
\tag{21}
$$

This emulates the effect of the dressed refocusing (phase) pulse. With this choice, the evolution consists of two periods with time-independent Hamiltonians, which is straight-forward to calculate on a computer.

In our implementation, which is available online, the parameters $\Omega_1$, $\Omega_2$, $r$, and $\theta$ are drawn in Monte-Carlo fashion from their respective distributions (Gaussian for the first three, $P(\theta) = \sin(\theta)$ with $0 \leq \theta \leq \pi/2$ for the latter). Statistical independence of the parameters is assumed. It is not unlikely that this assumption is at least partially wrong, since the respective orientation of the trityl moieties is restricted by the rigid linker. Although we have implemented simulations with a distance distribution, we do not consider such cases here but rather assume fixed values of $r$. Additionally, all the simulations shown

here assume on-resonance irradiation in the sense that the mean values of $\Omega_1$ and $\Omega_2$ are 0.

Some illustrative simulations are shown in Fig. 6. For each parameter set, we display the numerical simulation in time and frequency domain as solid lines and show the analytical dipolar powder pattern (scaled by 3/4) as dashed lines on top. In panel (a), we show simulations assuming infinitely narrow EPR lines. In this case, the numerical and analytical results are the



same. Panel (b) shows a simulation were we assume a FWHM of 16 MHz for both offset distributions (denoted by $\Gamma_\Omega$). For
the case of $r = 5.3$ nm, the simulation qualitatively reproduces the experimental results for bis-trityl **2**, especially regarding
the singularities in the dipolar spectra. For $r = 4.1$ nm, the experimental results actually look better than the simulation if one
regards the additional singularities at $3/8 \cdot d$ as an artifact. In this case, simulations with $\Gamma_\Omega = 8$ MHz are actually closer to
the experimental results (see panel (c)). This might suggest that the difference in offsets of bis-trityl **1** is smaller than the EPR
spectrum might suggest. Either hyperfine and dipolar couplings significantly contribute to the linewidth of the EPR spectrum, or
the offsets are not completely uncorrelated in reality. In order to guide future developments, we also simulated traces assuming

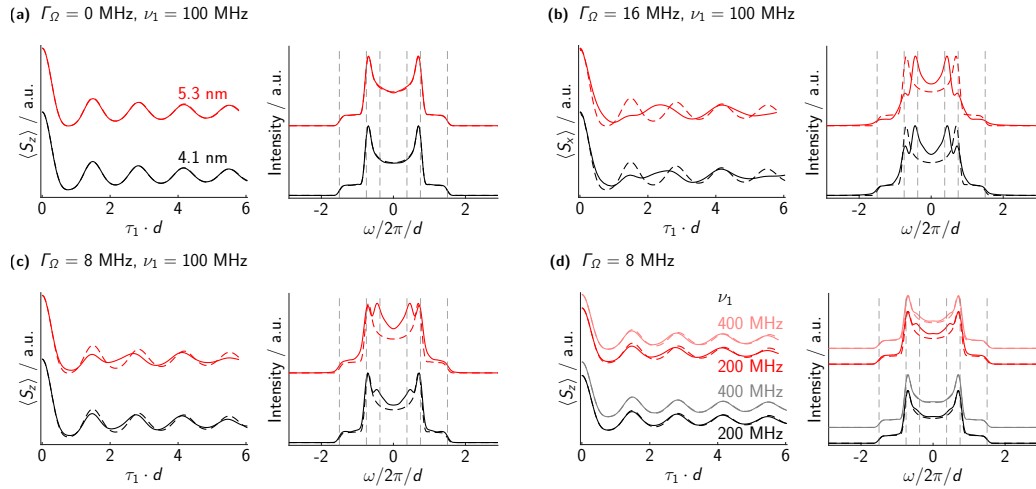

**Figure 6.** Numerical simulations with different dipolar couplings, offset distributions, and Rabi frequencies. Time and frequency axes are
scaled by the dipolar coupling to facilitate comparison. Dashed lines represent the analytical Pake pattern (with the frequency scaled by 3/4).
**(a)** No offsets at all, 100 MHz Rabi frequency. The numerical simulation of the spin-lock completely matches the analytical expectation.
**(b)** Gaussian offset distribution with FWHM of 16 MHz, 100 MHz Rabi frequency. The numerical simulations deviate from the analytical
expectation. In the frequency domain, "artifacts" appear at lower frequencies, around $3/8 \cdot \nu_\perp$. **(c)** Same as **(b)**, but with reduced offset
FWHM of only 8 MHz. The intensity of the artifacts is reduced compared to larger offset distributions. **(d)** same as **(c)**, but with increased
Rabi frequencies. The intensity of the artifacts is again reduced compared to smaller Rabi frequencies.


$\Gamma_\Omega = 8$, but with significantly larger microwave strengths of $\nu_1$=200/400 MHz, see panel (d). Compared to panels (b) and (c),
these simulations already show much better defined dipolar spectra. In conclusion, the simulations confirm that at least some
of the artifacts in the experimental results are due to the finite size of the electron spin nutation frequency. The contribution of
the artifacts becomes larger for larger offset differences and smaller dipolar couplings.

## 5  Conclusions and outlook

We showed that it is possible to measure the dipolar coupling between trityl radicals during a spin-lock by using short intervals
of phase-modulations, *i.e.* by a *dressed*-spin echo generated with PM pulses. The relaxation during the spin lock is much slower





compared to a simple two-pulse echo decay. The phenomena can be conceptually understood by describing the spin-lock in a nutating frame and using average Hamiltonian theory. For an electron-electron distance of ≈4.1 nm, the experimental spectra

agree very well with the theoretical expectations that assume a microwave Rabi frequency much larger than all other interactions in the system. For a distance of ≈5.3 nm, additional singularities appear in the dipolar spectrum. While the spin dynamics underlying these additional contributions can be understood by numerical simulations, they might seriously complicate data analysis in terms of distance distributions and have to be addressed in the future, if the sequence should be used in application work. Additionally, we showed a profound difference between the longitudinal and transverse rotating frame relaxation times,

$T_{1\rho}$ and $T_{2\rho}$. In our case, the latter is much smaller than the former and unfortunately limits the distance measurements by the sequence introduced here. Preliminary results with the OX063 trityl and its partially deuterated analogue OX71 in different solvent compositions (not shown) revealed that even bare-spin relaxation at low temperatures and low concentrations is complicated to understand, let alone dressed-spin relaxation with characteristic times $T_{2\rho}$ and $T_{1\rho}$. We are planning to investigate this in more detail.

Since there are still significant artifacts present in the dipolar spectra when measuring longer distances, we refrained from a systematic analysis of signal-to-noise ratio and a comparison with existing pulse sequences.

Nevertheless, we are confident that the presented obstacles can be overcome. First, it might very well be possible to come up with a dressed pulse sequence that measures the dipolar coupling with an observation time limited by $T_{1\rho}$ instead of $T_{2\rho}$. This appears feasible because, unlike the sum of dressed spin polarizations of the two spins, their difference is affected by dipolar coupling.

This fact is used in cross-polarization in solid-state NMR and oscillatory behavior of magnetization transfer in the rotating frame has been studied in the context of heteronuclear correlation spectroscopy (Müller and Ernst, 1979). Second, the ratio of Rabi frequency to offsets could be reduced by going to a lower field. While in principle we could have done the experiments at X-band frequencies, our TWT in this range can only generate pulses of up tp 15 μs. The Rabi frequencies generated by our setup are already rather high ($\approx 100$ MHz compared to ≈50 MHz in most commercial setups), but several

groups around the world are working on micro resonators (Anders and Lips, 2019; Sidabras et al., 2019; Narkowicz et al., 2008; Blank et al., 2017), which generally give higher conversion factors and could be used to generate higher Rabi frequencies. If these difficulties can be overcome, pulse dressed electron paramagnetic resonance could significantly expand the measurable distance range, at least for trityl radicals.

*Code and data availability.* Experimental data, processing scripts, and simulation scripts in MATLAB are available online. DOI: 10.5281/zen-
odo.3703053

*Author contributions.* NW designed the research with input from GJ. HH synthesized the bis-trityl rulers under the supervision of AG. NW carried out all measurements, data analysis and simulations with help from AV. The EPR part of the manuscript was written by NW and edited by GJ and AG. The synthetic part was written by HH and AG.



*Competing interests.* The authors declare that they have no con-flict of interest.

*Acknowledgements.* This work was financed by ETH Zürich (grant ETH-48 16-1) and the Deutsche Forschungsgemeinschaft within SPP 1601 (GO 555/6-2). We thank Lukas Schreder, who carried out a research project in a very independent manner and sparked NW's interest in dressed electron spin resonance. We thank Jan Henrik Ardankjær-Larsen for providing OX063 and OX071 for preliminary relaxation studies. Matthias Ernst is acknowledged for helpful discussions and critically reading the theoretical part of the article.



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
