# Peer review of "Distance measurement between trityl radicals by pulse dressed electron paramagnetic resonance with phase modulation"

_Magnetic Resonance, 2020_

## Short Comment (SC1) · 24 Mar 2020

Very nice work! Well-discussed theory and explanation of the spin physics. I suggest adding a DQC/SIFTER of the model compounds in SI.

---

## Referee Comment (RC1) · Thomas Prisner (Referee) · 29 Mar 2020

The manuscript of Wili et al. describes a very new and exciting experiment to measure dipolar couplings between two trityl radicals at Q-band frequencies using spin-lock techniques and phase modulation two pulse echo of this 'dressed spin states' in the nutation frame. This work adds a new possibilities to prolong the observation time window for the measurement of dipolar couplings in EPR which could be useful to extend the distance range in the future. Despite the fact that the experimental problems seen (and described very clear and fair in the manuscript) do not allow routine application of this methods yet for long distances (where the dipolar coupling strength is much less than the inhomogenous linewidth) this very new approach is very interesting.

The experiments as well as the theoretical description and discussion is very good, the literature is cited appropriately and the existing problems with this very new approach – especially limitations with respect to Rabi oscillation frequency strength in comparison with the inhomogeneous linewidth – is fair and clear described. There are many interesting aspects in this work, as for example also the large difference between T1rho and T2rho, which will stimulate further work in this direction. I recommend publication of this very nice and innovative article.

Some small remarks:

- Line 26 there should be an 'for' instead of 'or'
- The exchange interaction is explicitly mentioned in the theoretical part; also the fact that in the interaction frame it might gain some additional importance because the Zeeman splitting and the linewidth offsets disappear. But then it is not mentioned any more. Of course the two model systems will not show such contributions, but maybe the authors have investigated potential effects of this theoretically? It would be nice to have a remark on this aspect in the discussion (or conclusion). As far as I see all trityls will be in the strong coupling regime, so the method could also work for shorter distances, where such interaction might play a role.
- It will be interesting to see what happens with the deuterated trityl radicals. As mentioned this will be something for a new publication and might shine some more light on the big differences between T1rho and T2rho. Also the T dependence of these rotating frame relaxation rates could be very interesting (also to further optimize the experiment)
- The exponential 'stretch' factors for the fits of T1rho and T2rho should also be given.
- The modulation amplitude aPM was set to 0.3 for the experiments and it is also mentioned in the manuscript that the theoretical modelling brakes down if this factor becomes too large. Can this be more quantified?
-

---

## Short Comment (SC2) · 30 Mar 2020

Very nice work !! I liked the SI with the detailed didactic (educational) explanation of the interaction frame transformations. It would be helpful for the reader if you would also give an explicit expression for the propagator representing the first PM pi/2 in the SI. It can be understood from eq. 15 but as you did go to basic explanations in the SI for other parts it will be great to add this as well.

————————————————

---

## Referee Comment (RC2) · Jack H. Freed (Referee) · 2 Apr 2020

Comments on 'Distance measurements between trityl radicals by pulse dressed electron paramagnetic resonance with phase modulation' by N. Wili, et al.

Peter Borbat and Jack H. Freed

The manuscript "Distance measurements between trityl radicals by pulse dressed electron paramagnetic resonance with phase modulation" by N. Wili, et al. describes a novel PDS method based on clever evolving dipolar coupling in the spin-locked state, thereby improving the distance range for trityl spin labels. The manuscript is generally

correct and should be published with the consideration of the comments that we made.

This work describes an interesting development as so far it has not been demonstrated in ESR that spin-locked electron spins could be used to evolve selectively electron-electron dipolar coupling. The manuscript is well organized and clearly written. Also, extensive effort to synthesize and characterize rigid trityl biradicals and to simulate the evolution of coherence is shown. In particular, we like the implementation of the pulse sequence refocusing the nutation phase and providing the dipolar evolution sandwich in the locked state followed by the readout sequence, which can in principle exclude unwanted dipolar evolution. A simple two-pulse echo sequence used for readout appears sufficient at this stage, and by using SIFTER and DQC it would be possible to approach the somewhat shorter distance range. However, the dead-time cannot be excluded. Maybe the authors need to apply SIFTER or DQC sequences in the locked state.

It is encouraging that despite all the limitations imposed by the interaction strengths, the method does work and notably improves the dipolar evolution time for trityls. This work suggests a set of possible factors limiting T2(rho), and hopefully future work may be able to mitigate the effects of some of these factors. We doubt, however, that TWTA amplitude or phase noise contributes to shorter T2(rho). This is certainly not the case for amplitude noise, which within the locking bandwidth is estimated to be in the $\sim$1-10 mW range or maybe even less for a typical tube (less than -10 dBm/MHz noise spectral density). The phase noise of a TWTA (which is likely made by Applied System Engineering) is expected to be rather low. At least, the phase-noise test data for all amplifiers built over 20 years for ACERT supports this notion. Introducing phase noise, while possible, would be a complicated matter. AM/PM conversion in a saturated tube may be a possible way to test the effect on T1(rho) and T2(rho) to provide some insights on the instrumentation-imposed limits.

Unfortunately, the work gives no clue regarding what to expect at a different temperature for T1(rho) and T2(rho). The experimental setup allows for easy temperature

change and it is highly desirable to see the ratio of Tm/T2(rho) for at least one more temperature.

There is no comparison with the existing pulse sequences such as DQC or SIFTER, but we agree this may be unnecessary for this work. There is sufficient data in the literature for them, and PDS heavily relies on nitroxides anyway.

After emphasizing the power of major PDS methods in the introduction, the authors demonstrated that the sequence works at least for trityl labels, but they make no comment on whether it could be extended to any other known spin-label. The labels that are commonly used in biomedical research and are subjects of most of the key works cited are nitroxides, which demonstrate Tm's longer than the trityl's T2(rho) of this work. It is well known in this field that nitroxide labels quite often destabilize and precipitate proteins, the issue being even more critical with trityl labels, which are by no means mainstream. This is a significant limitation to the scope of this complex novel PDS method. Assuming that trityl labels were to have progressed to comparable use, there are other challenges that need to be addressed. The protein and lipid dynamics leading to Tm's in the low microsecond range as well as high local concentrations in the case of membrane proteins may contribute a set of problems in achieving T2ïĄš anywhere close to that observed in this work in dOTP glass.

Note that the T2(rho)'s obtained are considerably shorter than for nitroxides in this glass. We have (unpublished) data that demonstrate 40 $\mu$s evolution time in this glass using the DEER-5 method; 4-pulse DEER is also not very far from this mark. We also found very long Tm's for (partly) deuterated proteins (Georgieva et al., J. Biol. Chem., 2010). We think this work should be cited in the context of deuterated proteins.

The special technical requirements of this experiment to provide long intense locking pulses need to be described in greater detail. Such pulses are not normally used in pulse ESR. A 150 W TWTA was employed to achieve ~100 MHz Rabi frequency (36 G B1) and this power could last for about 40 $\mu$s periods limited by the amplifier. In the first

place it is a lot of power that can cause heating, arcing, and a damage to the receiver, thus limiting the repetition frequency. How was the receiver protected?

The origins of the baseline and of modulation depth need to be discussed.

Figure S8 – legends need be corrected.

———————————————

---

## Author Comment (AC1) · 14 Apr 2020

Thank you for your positive comments and suggestion to add SIFTER data in the SI. We now provide the data in the updated SI (attached with highlighted changes). Note that we would still like to refrain from a systematic and quantitative comparison between our sequence and SIFTER.

There might be additional changes in the SI in the final revised manuscript.

Please also note the supplement to this comment:
https://www.magn-reson-discuss.net/mr-2020-7/mr-2020-7-AC1-supplement.pdf

---

## Author Comment (AC2) · 14 Apr 2020

**Supporting Information - Part A**
**for**
**Distance measurement between trityl radicals by pulse dressed electron paramagnetic resonance with phase modulation**

Nino Wili[1], Henrik Hintz[2], Agathe Vanas[1,‡], Adelheidt Godt[2], and Gunnar Jeschke[2]

[1] *Department of Chemistry and Applied Biosciences, Laboratory of Physical Chemistry, ETH Zurich, Vladimir-Prelog-Weg 2, 8093 Zurich, Switzerland. E-mail: gjeschke@ethz.ch, nwili@ethz.ch*

[1] *Faculty of Chemistry and Center for Molecular Materials (CM$_2$), Bielefeld University, Universitätsstrasse 25, 33615 Bielefeld, Germany*

**Contents**

**S.1 Derivation of the nutating frame Hamiltonians**

In this section we will show step-by-step how to derive the effective Hamiltonians in the nutating frame.

**S.1.1 Interaction frame transformation**

We can split a Hamiltonian $\hat{\mathcal{H}}$ arbitrarily into a dominant part $\hat{\mathcal{H}}_0$ and all the other terms, $\hat{\mathcal{H}}_1$.

$$\hat{\mathcal{H}} = \hat{\mathcal{H}}_0 + \hat{\mathcal{H}}_1 \quad . \tag{1}$$

We can then transform $\hat{\mathcal{H}}$ into an interaction frame with $\hat{\mathcal{H}}_0$ with the unitary transformation

$$\hat{U} = \hat{T} \exp\left(-i \int_0^t \hat{\mathcal{H}}_0(t_1) \mathrm{d}t_1\right)$$

$$\hat{\mathcal{H}}' = \hat{U}^\dagger \hat{\mathcal{H}} \hat{U} \quad , \tag{2}$$

where $\hat{T}$ is the Dyson time-ordering operator and $\dagger$ indicates the transpose and complex conjugate. For a time-independent $\hat{\mathcal{H}}_0$, the expression simplifies to

$$\hat{U} = \exp\left(-i\hat{\mathcal{H}}_0 t\right) \quad . \tag{3}$$

The equation of motion of the density operator in the interaction frame is then given by

$$\frac{\mathrm{d}}{\mathrm{d}t} \rho' = -i\left[\hat{\mathcal{H}}' - \hat{\mathcal{H}}_0', \rho'\right] \tag{4}$$

$$\hat{\mathcal{H}}' - \hat{\mathcal{H}}_0' = \hat{\mathcal{H}}_1' \tag{5}$$

*i.e.* the influence of $\hat{\mathcal{H}}_0$ is absorbed into the frame, and we only have to look at the non-dominant terms in the interaction frame. Most of these calculations only involve simple rotations in 3-dimensional space.

**S.1.2 Rotating frame**

For simplicity and illustration, we first look at an isolated spin-1/2 particle in a static magnetic field $B_0$ along the laboratory $z$-axis. If we irradiate this system with a linearly polarized electromagnetic field with frequency $\omega_\mathrm{mw}$ and amplitude $2B_1$, the Hamiltonian in angular frequency units is given by

$$\hat{\mathcal{H}} = \omega_S \hat{S}_z + 2\omega_1 \cos\left(\omega_\mathrm{mw} t + \phi_\mathrm{mw}\right) \hat{S}_x \quad , \tag{6}$$

We now go into a rotating frame with frequency $\omega_{\text{mw}}$. This corresponds to an interaction frame with $\omega_{\text{mw}}\hat{S}_z$, i.e. $\hat{\mathcal{H}}_0 = \omega_{\text{mw}}\hat{S}_z$, which leads to

$$
\begin{aligned}
\hat{\mathcal{H}}_1' =\,& \omega_S \hat{S}_z \\
& + 2\omega_1 \cos\left(\omega_{\text{mw}}t + \phi_{\text{mw}}\right)\left(\cos(\omega_{\text{mw}}t)\hat{S}_x - \sin(\omega_{\text{mw}}t)\hat{S}_y\right) \\
& - \omega_{\text{mw}}\hat{S}_z \\
=\,& \Omega_S \hat{S}_z \\
& + \omega_1\left(\cos(\phi_{\text{mw}})\hat{S}_x + \sin(\phi_{\text{mw}})\hat{S}_y\right) \\
& + \omega_1\left(\cos(2\omega_{\text{mw}}t + \phi_{\text{mw}})\hat{S}_x - \sin(2\omega_{\text{mw}}t + \phi_{\text{mw}})\hat{S}_y\right) \quad,
\end{aligned}
\tag{7}
$$

where we used trigonometric identities and introduced the offset $\Omega_S = (\omega_S - \omega_{\text{mw}})$. Equation (7) contains two rotating waves, one that rotates with $\omega_{\text{mw}}$ that is said to be *on-resonance*, and one that rotates with $-\omega_{\text{mw}}$ and is thus far *off*-resonance. The rotating wave approximation (RWA) now simply neglects the counter-rotating component. More formally speaking, this corresponds to the first order average Hamiltonian, obtained by integrating over one modulation period, in this case $\tau_c = 2\pi/\omega_{\text{mw}}$,

$$
\hat{\overline{\mathcal{H}}}^{(1)} = \frac{1}{\tau_c}\int_0^{\tau_c}\hat{\mathcal{H}}(t_1)\mathrm{d}t_1 \quad.
\tag{8}
$$

Here, the bar indicates the average Hamiltonian, the number in parentheses indicates the order. Quite often, and also in the case of the RWA, this amounts to simply neglecting the remaining time-dependent terms, as they are off-resonant. For the Hamiltonian in Eq. (7) we obtain

$$
\hat{\mathcal{H}}_1' \approx \Omega_S \hat{S}_z + \omega_1\left(\cos(\phi_{\text{mw}})\hat{S}_x + \sin(\phi_{\text{mw}})\hat{S}_y\right)
\tag{9}
$$

The second order contribution to the average Hamiltonian is given by

$$
\hat{\overline{\mathcal{H}}}^{(2)} = \frac{-i}{2\tau_c}\int_0^{\tau_c}\int_0^{t_2}\left[\hat{\mathcal{H}}(t_2),\hat{\mathcal{H}}(t_1)\right]\mathrm{d}t_1\mathrm{d}t_2 \quad.
\tag{10}
$$

Evaluating the double-integral for Eq. (7) leads to the Bloch-Siegert shift Hamiltonian

$$
\hat{\mathcal{H}}_{\text{BS}} = \frac{1}{4}\frac{\omega_1^2}{\omega_{\text{mw}}}\hat{S}_z \quad.
\tag{11}
$$

**S.1.3  Nutating Frame**

The truncated rotating frame Hamiltonian in Eq. (9) can be further simplified by assuming $\phi_{\text{mw}} = 0$, which leaves us with

$$
\hat{\mathcal{H}}' = \Omega_S \hat{S}_z + \omega_1 \hat{S}_x
\tag{12}
$$

If the Rabi frequency is much larger than the resonance offset, $\omega_1 \gg \Omega_S$, the situation can be discussed by using another interaction frame transformation with $\omega_1 \hat{S}_x$. This term is then absorbed into the frame, and the rest transforms to

$$\hat{\mathcal{H}}'' = \Omega_S \left( \cos(\omega_1 t) \hat{S}_z + \sin(\omega_1 t) \hat{S}_y \right) \quad . \tag{13}$$

Here, all terms are time-dependent. Accordingly, the first order average Hamiltonian vanishes

$$\hat{\bar{\mathcal{H}}}''^{(1)} = 0 \quad . \tag{14}$$

It is instructive to look at the second order contribution, which amounts to

$$\hat{\bar{\mathcal{H}}}''^{(2)} = \frac{\Omega_S^2}{2\omega_1} \hat{S}_x \quad . \tag{15}$$

These results can be interpreted in the following way. To a first approximation, the offsets in the rotating frame get averaged out to zero. If this approximation is good, then electrons with different resonance frequencies in the rotating frame have the same resonance frequency in the nutating frame. They become equivalent under the spin-lock. As a second approximation, there is a correction that scales with $\Omega_S^2/\omega_1$, and this correction acts along the spin-lock axis of the nutating frame. It depends on the bare-spin resonance offset, $\Omega_S$, and it will thus be different for electrons with different bare-spin resonance frequencies. If the bare-spin resonance offset becomes comparable to the Rabi frequency, it is no longer permissible to neglect higher-order terms, and it is better to use an interaction frame with $\Omega_S \hat{S}_z + \omega_1 \hat{S}_x$. This corresponds to a nutating frame where the axis of rotation is neither along $z$ nor along $x$, but somewhere in between. However, now the frame transformation depends on the bare-spin resonance frequency, *i.e.*, on a parameter of the spin system rather than only on parameters chosen by the experimenter and being the same for all spins.

**S.1.3.1   Hyperfine decoupling and NOVEL**

The Hamiltonian for a single electron coupled to a single nucleus in the electron rotating frame is given by

$$\hat{\mathcal{H}}'_{\text{e-n}} = \hat{S}_z \left( A_{xz} \hat{I}_x + A_{yz} \hat{I}_y + A_{zz} \hat{I}_z \right) + \omega_I \hat{I}_z \quad . \tag{16}$$

An interaction frame transformation with $\omega_1 \hat{S}_x$ leads to

$$\hat{\mathcal{H}}''_{\text{e-n}} = \left( \cos(\omega_1 t) \hat{S}_z + \sin(\omega_1 t) \hat{S}_y \right)$$
$$\cdot \left( A_{xz} \hat{I}_x + A_{yz} \hat{I}_y + A_{zz} \hat{I}_z \right) + \omega_I \hat{I}_z \quad . \tag{17}$$

Again, all terms are time-dependent and the first order average Hamiltonian vanishes. If the dominant electron dephasing mechanism relies on the hyperfine coupling, then the dephasing during irradiation should be strongly reduced.

There is, however, a hidden complication, if $\omega_1 \approx \omega_I$. This is best seen by going into the nuclear

rotating frame, *i.e.* the interaction frame with $\omega_I \hat{I}_z$. This leads to

$$
\hat{\mathcal{H}}''_{\text{e-n}} = \left( \cos(\omega_1 t) \hat{S}_z + \sin(\omega_1 t) \hat{S}_y \right)
$$
$$
\cdot \left[ A_{xz} \left( \cos(\omega_I t) \hat{I}_x - \sin(\omega_I t) \hat{I}_y \right) \right.
$$
$$
+ A_{yz} \left( \cos(\omega_I t) \hat{I}_y + \sin(\omega_I t) \hat{I}_x \right)
$$
$$
\left. + A_{zz} \hat{I}_z \right] \quad . \tag{18}
$$

If $\omega_1$ matches $\omega_I$, some terms proportional to the pseudo-secular couplings, $A_{xz}$ and $A_{yz}$, become time-independent. Using trigonometric identities and neglecting time-dependent terms, we obtain

$$
\hat{\mathcal{H}}''_{\text{e-n}} \approx \frac{1}{2} \left[ \hat{S}_y \left( A_{yz} \hat{I}_x - A_{xz} \hat{I}_y \right) \right.
$$
$$
\left. + \hat{S}_z \left( A_{xz} \hat{I}_x + A_{yz} \hat{I}_y \right) \right] \quad . \tag{19}
$$

This Hamiltonian generates nuclear polarization ($\hat{I}_z$) from electron coherence ($\hat{S}_x$) and is thus a mechanism for dynamic nuclear polarization (DNP). The matching condition $\omega_1 = \omega_I$ is called the NOVEL condition[2]. In order to avoid magnetization loss, microwave field strengths close to the NOVEL condition should be avoided unless such polarization transfer is intended.

**S.1.3.2   Qualitative discussion of electron dephasing in a nuclear spin bath**

Let us look at an electron spin in a bath of nuclei. For simplicity and illustration, we only consider one electron and two nuclei. We assume that the electron is only coupled to nucleus one, we neglect the pseudosecular components of the hyperfine coupling and we do include dipole-dipole coupling between the two nuclei. The Hamiltonian in the electron rotating frame thus reads

$$
\hat{\mathcal{H}}' = A_{zz} \hat{S}_z \hat{I}_z + \omega_{12} \left( \hat{I}_{1z} \hat{I}_{2z} - \frac{1}{2} \left( \hat{I}_{1x} \hat{I}_{2x} + \hat{I}_{1y} \hat{I}_{2y} \right) \right) \quad , \tag{20}
$$

where $\omega_{12}$ is the dipolar coupling constant of the two nuclei (and thus depends on the distance between the nuclei and on their gyromagnetic ratios). Let us assume that we created electron coherence along $x'$ and that it further evolves under the sequence $\tau - (\pi)_y - \tau$. If $\omega_{12} = 0$, the $\pi$-pulse completely refocuses the hyperfine coupling, and we end up with $-\hat{S}_{x'}$ after the second delay. However, as the flip-flop terms in the nuclear-nuclear coupling Hamiltonian do not commute with the other terms, for $\omega_{12} \neq 0$ evolution generates two- and three-spin terms of the form $\hat{S}_y \hat{I}_{1x} \hat{I}_{2y}$ (among others). The full expression is rather large and does not provide much insight. However, one easily recognizes that these terms are not refocused by a microwave $\pi$-pulse acting on the electron spin. Since the terms are not refocused, the amplitude of the electron coherence must necessarily be reduced. Note that this effect is completely coherent. It does not rely on stochastic flips of the nuclear spins. Extending the argument to a large number of nuclear spins with a distribution of hyperfine and nuclear-nuclear couplings, one expects an apparent decay instead of oscillations. A demonstration by numerical simulations would be computationally expensive, as matrix size grows exponentially with the number of spins. Such computations are beyond the scope of this paper.

Note, however, that a spin-lock that is much stronger than the hyperfine coupling averages said coupling. The nuclear-nuclear coupling can then be neglected because it commutes with all the remaining terms that contain electron operators. In other words, a sufficiently strong spin lock isolates the electron spin from the nuclear spin bath, unless the NOVEL condition is met.

**S.1.3.3 Electron-electron coupling in the nutating frame**

The dipolar coupling Hamiltonian between two electron spins in the electron rotating frame is given by

$$\hat{\mathcal{H}}'_{\text{e-e,dip}} = \omega_{\text{dd}} \left( \hat{S}_{1z}\hat{S}_{2z} - \frac{1}{2} \left( \hat{S}_{1x}\hat{S}_{2x} + \hat{S}_{1y}\hat{S}_{2y} \right) \right) \quad , \tag{21}$$

as already mentioned in the main text. The situation of strong irradiation of *both* spins is well pictured in a nutating frame obtained by transformation with $\omega_1(\hat{S}_{1x} + \hat{S}_{2x})$. This leads to

$$\hat{\mathcal{H}}''_{\text{e-e,dip}} = \omega_{\text{dd}} \left[ (c\hat{S}_{1z} + s\hat{S}_{1y})(c\hat{S}_{2z} + s\hat{S}_{2y}) \right.$$

$$\left. - \frac{1}{2} \left( \hat{S}_{1x}\hat{S}_{2x} + (c\hat{S}_{1y} - s\hat{S}_{1z})(c\hat{S}_{2y} - s\hat{S}_{2z}) \right) \right] \tag{22}$$

with

$$c = \cos(\omega_1 t)$$

$$s = \sin(\omega_1 t) \quad . \tag{23}$$

Using the first order average Hamiltonian approximation and the integrals

$$\frac{1}{\tau_c} \int_0^{\tau_c} c \cdot s \, dt = 0 \tag{24}$$

$$\frac{1}{\tau_c} \int_0^{\tau_c} c^2 dt = \frac{1}{\tau_c} \int_0^{\tau_c} s^2 dt = 1/2 \tag{25}$$

$$\tag{26}$$

we obtain the result shown in the main text

$$\hat{\mathcal{H}}''_{\text{e-e,dip}} \approx \omega_{\text{dd}} \left[ \frac{1}{2}\hat{S}_{1z}\hat{S}_{2z} + \frac{1}{2}\hat{S}_{1y}\hat{S}_{2y} \right.$$

$$\left. - \frac{1}{2} \left( \hat{S}_{1x}\hat{S}_{2x} + \frac{1}{2}\hat{S}_{1y}\hat{S}_{2y} + \frac{1}{2}\hat{S}_{1z}\hat{S}_{2z} \right) \right]$$

$$= -\frac{\omega_{\text{dd}}}{2} \left[ \hat{S}_{1x}\hat{S}_{2x} - \frac{1}{2} \left( \hat{S}_{1z}\hat{S}_{2z} + \hat{S}_{1y}\hat{S}_{2y} \right) \right] \tag{27}$$

Note that the simplification leading to this result crucially depends on the condition that both spins are irradiated with the same strength and that the irradiation is stronger than the difference in rotating frame offsets. If the difference in offsets is larger than the dipolar coupling or the Rabi frequency,

$\Delta\Omega > \omega_1, \omega_{dd}$, then the pseudosecular terms in Eq. (21) are already averaged in the doubly rotating frame (meaning that the frames for the two electrons rotate with different frequencies). If only one spin is irradiated, the average dipole-dipole coupling Hamiltonian will vanish. However, if the two spins were irradiated separately with two different frequencies and the same Rabi frequency, part of the Hamiltonian would again be time-independent. This is a situation very similar to Hartmann-Hahn cross-polarization in solid-state NMR.

Regarding the exchange coupling, one can simply argue that it involves a scalar product of spin operators. The scalar product is invariant under frame transformations. However, this statement is slightly dubious because the scalar product of spin operators connects two different spin spaces and may no longer be invariant if the transformations of the two frames differ. Therefore, depending on the irradiation scheme the exchange coupling is not trivially the same between dressed spins as between bare spins. Here, we show explicitly that the exchange coupling is invariant if the same frame transformation is applied to both spins. For an exchange coupling Hamiltonian given by

$$
\begin{aligned}
\hat{\mathcal{H}}'_{\text{e-e,J}} &= J\left(\hat{\vec{S}}_1 \cdot \hat{\vec{S}}_2\right) \\
&= J\left(\hat{S}_{1x}\hat{S}_{2x} + \hat{S}_{1y'}\hat{S}_y + \hat{S}_{1z}\hat{S}_{2z}\right) \quad,
\end{aligned}
\tag{28}
$$

we employ an interaction frame transformation with $\omega_1(\hat{S}_{1x} + \hat{S}_{2x})$, leading to

$$
\begin{aligned}
\hat{\mathcal{H}}''_{\text{e-e,J}} &= J\,(\hat{S}_{1x}\hat{S}_{2x} \\
&\quad + (c\hat{S}_{1y} - s\hat{S}_{1z})(c\hat{S}_{2y} - s\hat{S}_{2z}) \\
&\quad + (c\hat{S}_{1z} + s\hat{S}_{1y})(c\hat{S}_{2z} + s\hat{S}_{2y})) \\
&= J\left(\hat{S}_{1x}\hat{S}_{2x} + \hat{S}_{1y}\hat{S}_{2y} + \hat{S}_{1z}\hat{S}_{2z}\right) \quad.
\end{aligned}
\tag{29}
$$

which is the same as before the interaction frame transformation.

**S.1.3.4   Details on the dressed spin echo**

Here we explicitly show some more details regarding the dressed spin echo sequence. We assume that the electron spins as well as the spin-lock are along the $x'$ axis. The density operator in the usual rotating frame is then described by

$$
\hat{\sigma}'_1 = \hat{S}_{1x} + \hat{S}_{2x} \quad.
\tag{30}
$$

The Hamiltonian describing the microwave irradiation in the rotating frame is given by

$$
\hat{\mathcal{H}}'_{\text{mw}} = \omega_1\left(\cos\left(\phi_{\text{mw}}(t)\right)\hat{F}_x + \sin\left(\phi_{\text{mw}}(t)\right)\hat{F}_y\right) \quad,
\tag{31}
$$

with $\hat{F}_x = \hat{S}_{1x} + \hat{S}_{2x}$. The above expression, in the absence of any phase-modulation ($\phi_{\text{mw}}(t) = 0$), reduces simply to

$$
\hat{\mathcal{H}}'_{\text{mw}} = \omega_1\hat{F}_x \quad.
\tag{32}
$$

As long as the spins and the spin-lock field point along the same direction, nothing happens (apart from relaxation, which is not treated in this formalism). This is even true if we include the dipolar coupling, because again, the sum of polarization does not evolve under the dipolar coupling Hamiltonian.

In order to describe a phase pulse in the rotating frame, we would need to use the Hamiltonian in Eq. (31). The corresponding propagator is given by

$$\hat{U} = \hat{T} \exp\left(-i \int_0^{t_p} \hat{\mathcal{H}}'_{\mathrm{mw}}(t) \mathrm{d}t\right) \tag{33}$$

where $\hat{T}$ is the Dyson time ordering operator. This integral is more of a formality, because it is usually not evaluated this way. Numerically, one can calculate the propagator by time-slicing, *i.e.* by evaluating the time-dependent Hamiltonian and the corresponding propagator for short time steps and calculating the integral by multiplying propagators of the small time steps in the correct order. However, it is much easier to describe a phase pulse in the nutating frame! In this frame, the density matrix above is still given by

$$\hat{\sigma}''_1 = \hat{S}_{1x} + \hat{S}_{2x} \ , \tag{34}$$

but the Hamiltonian, after the rotating wave approximation, reduces to

$$\hat{\mathcal{H}}''_{\mathrm{mw}} = \frac{\omega_1 a_{\mathrm{PM}}}{2} \left(\cos\left(\phi_{\mathrm{PM}}\right) \hat{F}_y + \sin\left(\phi_{\mathrm{PM}}\right) \hat{F}_z\right) \ , \tag{35}$$

which for $\phi_{\mathrm{PM}} = \pi$ (arbitrary) reduces to

$$\hat{\mathcal{H}}''_{\mathrm{mw}} = -\frac{\omega_1 a_{\mathrm{PM}}}{2} \hat{F}_y \ . \tag{36}$$

This expression is time-independent. The corresponding propagator is then given by

$$\hat{U} = \exp\left(i \frac{\omega_1 a_{\mathrm{PM}}}{2} \hat{F}_y t_p\right) \ . \tag{37}$$

If we choose $\frac{\omega_1 a_{\mathrm{PM}}}{2} \cdot t_p = \pi/2$, we obtain

$$\hat{U} = \exp\left(i \frac{\pi}{2} \hat{F}_y\right) \ , \tag{38}$$

which corresponds to a $\frac{pi}{2}$ pulse in the nutating frame. Applied to the density matrix before the pulse, we get

$$\hat{\sigma}''_2 = \hat{U} \hat{\sigma}''_1 \hat{U}^\dagger = \hat{S}_{1z} + \hat{S}_{2z} \ . \tag{39}$$

The last line can be represented in the product operator formalism (POF) by

$$\hat{S}_{1x} + \hat{S}_{2x} \xrightarrow{-\frac{\pi}{2}\hat{F}_y} (\hat{S}_{1z} + \hat{S}_{2z}) \tag{40}$$

Now we explicitly show how this state evolves under the dipolar Hamiltonian in the nutating frame, *i.e.*

$$\hat{S}_{1z} + \hat{S}_{2z} \xrightarrow{-\frac{\omega_{dd}}{2}\left(\hat{S}_{1x}\hat{S}_{2x} - \frac{1}{2}\left(\hat{S}_{1z}\hat{S}_{2z} + \hat{S}_{1y}\hat{S}_{2y}\right)\right)\cdot t} ? \tag{41}$$

It is important to note that all the two-spin operators in the Hamiltonian commute with each other. This means that we can evaluate their influence on the density operator step-by-step in the POF. We also do it for one spin only and then obtain the result of the second spin by changing the corresponding indices.

$$\hat{S}_{1z} \xrightarrow{-\frac{\omega_{dd}}{2}\left(\hat{S}_{1x}\hat{S}_{2x}\right)\cdot t} \cos(\frac{\omega_{dd}}{4}t)\hat{S}_{1z} + \sin(\frac{\omega_{dd}}{4}t)\hat{S}_{1y}\hat{S}_{2x}$$

$$\xrightarrow{+\frac{\omega_{dd}}{4}\left(\hat{S}_{1y}\hat{S}_{2y} + \hat{S}_{1z}\hat{S}_{2z}\right)\cdot t} \cos(\frac{\omega_{dd}}{4}t)\left[\cos(\frac{\omega_{dd}}{8}t)\hat{S}_{1z} + \sin(\frac{\omega_{dd}}{8}t)\hat{S}_{1x}\hat{S}_{2z}\right]$$

$$+\sin(\frac{\omega_{dd}}{4}t)\left[\cos(\frac{\omega_{dd}}{8}t)\hat{S}_{1y}\hat{S}_{2x} - \sin(\frac{\omega_{dd}}{8}t)\hat{S}_{2z}\right] \tag{42}$$

Including the result for the second spin, and only keeping terms proportional to $\hat{S}_{1z} + \hat{S}_{2z}$ gives the signal

$$s(t) = \cos(\frac{\omega_{dd}}{4}t)\cos(\frac{\omega_{dd}}{8}t) - \sin(\frac{\omega_{dd}}{4}t)\sin(\frac{\omega_{dd}}{8}t)$$

$$= \frac{1}{2}\left(\cos(\frac{3}{8}\omega_{dd}t) + \cos(\frac{1}{8}\omega_{dd}t) + \cos(\frac{3}{8}\omega_{dd}t) - \cos(\frac{1}{8}\omega_{dd}t)\right)$$

$$= \cos(\frac{3}{8}\omega_{dd}t) \tag{43}$$

Note that in the main text, we display the echo intensity as a function of $\tau_1$. The effective dipolar evolution time is $2 \cdot \tau_1$, leading to

$$s(t) = \cos(\frac{3}{4}\omega_{dd}\tau_1) \tag{44}$$

**S.2   Setting up the dressed spin echo sequence**

Here we give a detailed account of the measurement setup for dressed pulse sequences. We assume that the sample is in the spectrometer at the correct temperature and field and that you are able to get an echo.

1. **Set up $\pi/2$ and $\pi$ pulse lengths** by maximizing the echo amplitude of a Hahn echo where the second pulse is twice as long as the first one. This step seems trivial, but it already suffers from some complications in the case of dipolar coupled trityls, which are seen in the following Figure S1. Clearly, there are two close lying local maxima, which is not compatible with the behavior of a simple spin-1/2 system. The local maxima are due to the small ratio of spectral width compared to the dipolar coupling, and the exact curve depends on the value of the interpulse delay. Note that for the dressed spin sequences, one generally prefers to choose this interpulse delay for the readout as short as possible. A similar observation was reported by Meyer *et al.*[3]

[Figure]

Fig. S1: Echo intensity of a 4/8ns echo as a function of the digital scale of our AWG. (Bis-trityl **1**). Note that starting from around 0.3, the saturation of the TWT becomes significant.

2. **Set up the optimal phase of the spin-lock** with a sequence $\pi/2 - Lock - \tau - \pi - \tau$- echo. In principle, this should not be necessary when using an AWG. In fact, we checked that the phase cycling in our setup works nearly perfectly when simply programming the phases. Additionally, in a two-pulse echo, the phase of the signal perfectly follows the expected phase shift of the coherence order pathway. If this is the case, one would expect that one could simply apply the spin-lock pulse at 90 degrees to the $\pi/2$ pulse, but experimentally, we observe that this is not the case, see Fig. S2. We suspect that phase transients of the pulses due to the resonator are responsible for this behavior.

3. **Measure $T_{1\rho}$** with the sequence $\pi/2 - Lock - \tau - \pi - \tau$ -echo (SL + echo) with variable length of the spin-lock pulse. Note that a spin-locked echo ($\pi/2 - \tau - Lock - \tau$ - echo, SLE) gives very different results in the case of trityl radicals at low temperatures, see the following Fig. S3. Note also that in the case of the SLE, the decay depends on the interpulse delay. Curiously, one can again see that the maximal echo intensity is not achieved when the spin-lock pulse corresponds to a $\pi$ pulse, but when it corresponds to a $\pi/2$ or $3\pi/2$ pulse. The red traces in Fig.

[Figure]

Fig. S2: Echo intensity after a sequence $\pi/2 - Lock - \tau - \pi - \tau$- echo as a function of the digital phase of the spin-lock pulse relative to the first pulse. The length of the spin-lock pulse was $5\mu s$. The highest expected absolute intensity of the echo is at around 120 degrees, while one would expect it around 90 degrees.

S3 correspond to a nutation of the second pulse. Obviously, not only the fundamental oscillation with the Rabi frequency of slightly more than 100 MHz is present, but also an oscillation which is twice as fast. This is due to the fact that a significant part of the echo reduction of the two-pulse echo is due to the dipolar coupling, which is refocused by a solid echo, not a Hahn echo. The solid echo requires a $\pi/2$ pulse. Note however, that the solid echo does not fully refocus the dipolar coupling in the presence of significant offsets compared to the interpulse delay.

[Figure]

Fig. S3: Measurement of $T_{1\rho}$ with different sequences and different settings. See main text for more details. **(b)** is a zoom of **(a)**.

4. **Record a frequency-stepped dressed spin resonance spectrum.** In order to set the frequency of the phase-modulation, we need to know the Rabi frequency of the spin-lock pulse. It is also useful to know the distribution of the Rabi frequency. One possibility to obtain this information is to use the basic sequence introduced in the main text, but with only a single period of phase-modulation. The amplitude of the phase-modulation should be rather small (*e.g.* 0.02 radians), and the pulse should be rather long (*e.g.* 1 $\mu s$.) in order to avoid significant broadening. The

frequency of the phase-modulation is then stepped, and the echo intensity recorded. This is shown in black in Fig. S4. As a comparison, we also show the Fourier transform of a simple nutation experiment in red. The frequency of the phase-modulation for all successive steps was then set to the maximum of the dressed spin resonance spectrum.

[Figure]

Fig. S4: Frequency-stepped dressed-spin resonance spectrum (black, $a_{pm} = 0.02$, phase pulse length 1 $\mu$s). And Fourier transform of a nutation experiment at full power (red). Insets show the pulse sequences. PM denotes the phase modulation during the spin-lock, not an additional channel. Based on this spectrum, a PM frequency of $\omega_{PM}/(2\pi) = 108$ MHz was chosen.

5. **Set up the phase-modulation amplitude and phase-pulse length** Now that the frequency of the phase modulation ($\omega_{PM}$) is set, we can setup the phase-pulse amplitude and length. We can do this by using the same sequence as in the main text, but with only one phase-pulse with fixed modulation frequency. By changing the length of the phase-modulation pulse one obtains a nutation curve in the nutating frame. Fig. S5 shows such nutation curves for $a_{PM} = 0.2$ (grey and black) and $a_{PM} = 0.3$ (red and light red). Interestingly, one can see an overlaid oscillation, which depends on the phase of the phase modulation ($\phi_2$). This additional oscillation is due to the breakdown of the rotating wave approximation in the nutating frame and becomes more pronounced with larger phase-modulation intensity. In this case we used the minimum of the phase-pulse nutation curve as a dressed $\pi$-pulse length (around 43 ns for $a_{PM} = 0.3$).

6. **Set up the timings for the dressed echo and check the phase-phase cycling.** The sequence from the main text is repeated here in Fig. S6 (a). The dressed echo cannot be acquired continuously. Therefore, we need to check whether the timings of the sequence are correct. In order to do this, we use a fixed spin-lock length $T_{SL}$, a fixed $\tau_0$ and $\tau_1$ and changed $\tau_2$ step-by-step. A trace with a two-step phase-cycle of $\phi_1 = [(+)0, (-)180]$ and $\phi_2 = \phi_3 = \phi_4 = 0$ is shown in red in Fig. S6 (b). The highest intensity indicates the dressed echo where $\tau_2 = \tau_1$ (plus the length of the first phase pulse, $t_p$). In addition, one can clearly see additional, *crossing* echoes, and a "dressed FID" around $\tau_2 = 0$. We realized that the position of these crossing echoes depend on the choice of $\tau_0$. This indicates that there is some dressed coherence present during $\tau_0$. This is not very surprising, because the first $\pi/2$ pulse will never be perfect, and some magnetization will be orthogonal to the effective field of the spin-lock. In order to get rid of these crossing echoes, one can cycle the phase of the phase-pulses $\phi_{2-4}$ in analogy to a bare spin echo, with the complication that at the beginning of the first dressed pulse (after $\tau_0$),

[Figure]

Fig. S5: **(a)** Pulse sequence to set up the dressed pulses. "PM" denotes the phase-modulation during the spin-lock, *not* and additional channel. **(b)** Phase-pulse nutation curves obtained with fixed $\omega_{PM}$ but different intensities and phases. The minima of the curves indicate the length of the dressed $\pi$ pulses for a given intensity $a_{PM}$.

there is already dressed coherence present. The black trace in Fig. S6 (b) shows the result of the measurement with the 16-step nested phase cycle $\phi_1 = [(+)0, (-)180]$, $\phi_2 = [(+)0, (-)180]$, $\phi_3 = [(+)0, (-)90, (+)180, (-)270]$, which essentially gets rid of all crossing echoes observed in our case. Note that this cycle relies on the fact that dressed coherence decays relatively quickly after the last phase-pulse due to inhomogeneities, and that it does not get refocused after that point. The exact implementation of the phase-modulation and thus the meaning of the phases are given in the next section.

**(a)**

[Figure]

**(b)**

[Figure]

Fig. S6: **(a)** Pulse sequence of the dressed spin echo. **(b)** Dressed echo with fixed $\tau_0 = 1\mu s$ and $\tau_1 = 1.5\mu s$ and variable $\tau_2$. Red: $\phi_1 = [(+)0, (-)180]$. Clear crossing echoes are visible. Black: Nested cycle $\phi_1 = [(+)0, (-)180]$, $\phi_2 = [(+)0, (-)180]$, $\phi_3 = [(+)0, (-)90, (+)180, (-)270]$. No crossing echoes visible anymore.

**S.3 Implementation details**

On our homebuilt AWG based spectrometer, the actual pulse waveform that reaches the sample is generated by IQ-mixing the AWG waveform (IF) with a local oscillator (LO). As an example, we use a Q-band phase-modulated pulse. The LO in this case is around 33.3 GHz. The center frequency of the pulse generated by the AWG is always around 1.5 GHz in our setup. This leads to a final pulse frequency of 33.3+1.5 = 34.8 GHz that reaches the resonator and the spins, as seen in Fig. 3(a), 4(a) and 5(a) in the main text. Apart from the constant value of around 1.5 GHz, we do not use any frequency modulation (FM) in this work ($FM(t) = 1.5$ GHz). The amplitude is also constant during the spin-lock. So let us look now at the phase modulation (PM), which is illustrated in Fig. S7 in red. At most times during the spin-lock, there is no phase modulation. But during the "phase pulses", we turn on the sinusoidal PM. It is best to think about these phase pulses in relation to a hypothetical "carrier". The frequency of this carrier is the dressed spin resonance frequency $\omega_{\mathrm{PM}} \approx \omega_1$ that we determined before in Fig. S4. The phase of this carrier is arbitrarily set to 0 at the beginning of the spin-lock pulse. In our experiment, all three phase pulses have the same frequency as the carrier, but each phase pulse might be phase shifted. This is seen in the example below. The first pulse has a phase $\phi_2 = 0$, which means that the phase is the same for the hypothetical carrier and the phase pulse. The second phase pulse is phase shifted by $\phi_3 = \pi$, and the third one by $\phi_4 = \pi/2$. By cycling the phases of the phase pulses, we can select certain dressed coherence order pathways, in analogy to the bare spin case. Of course the amplitude of the phase modulation (0.3 rad in the example shown here) as

[Figure]

Fig. S7: Illustration of the

well as the length of each phase pulse can be chosen at will. Mathematically, the PM is given by

$$
PM(t) = \begin{cases}
a_{\mathrm{PM},1} \cos\left(\omega_{\mathrm{PM}}t + \phi_2\right) & \text{for } \tau_0 < t < \tau_0 + t_{\mathrm{PM},1} \\
a_{\mathrm{PM},2} \cos\left(\omega_{\mathrm{PM}}t + \phi_3\right) & \text{for } (\tau_0 + \tau_1) < t < (\tau_0 + \tau_1 + t_{\mathrm{PM},2}) \\
a_{\mathrm{PM},3} \cos\left(\omega_{\mathrm{PM}}t + \phi_4\right) & \text{for } (\tau_0 + \tau_1 + \tau_2) < t < (\tau_0 + \tau_1 + \tau_2 + t_{\mathrm{PM},3}) \\
0 & \text{otherwise}
\end{cases}
\tag{45}
$$

The total final spin-lock waveform generated by the two channels of the AWG (real and imaginary) is then given by

$$
\begin{aligned}
f(t) = AM(t) \cdot \Bigg[ &\cos\left( 2\pi \cdot \int_0^t FM(\tau)\mathrm{d}\tau + PM(t) + \phi_{\mathrm{SL}} \right) \\
&+ i\sin\left( 2\pi \cdot \int_0^t FM(\tau)\mathrm{d}\tau + PM(t) + \phi_{\mathrm{SL}} \right) \Bigg] \quad,
\end{aligned}
\tag{46}
$$

where $\phi_{\mathrm{SL}}$ is the overall phase of the spin-lock pulse. We omitted the possible corrections for phase and amplitude for the real and imaginary part.

**S.4  Additional experimental data**

**S.4.1  Dipolar echo envelope modulation: Influence of pulse length**

As mentioned in the main text, the dipolar coupling is already visible in the normal Hahn echo decay. Fig. S8 shows these echo decays for the two different bis-trityl compounds, and with different pulse lengths. For both bis-trityls, we see and additional oscillation when very short pulses (4/8 ns) are used. With 100/200 ns pulses, the modulation intensity is reduced. In the case of bis-trityl **2** (5.3 nm) and 4/8 ns pulses, the echo modulation even crosses zero. The presence of these oscillations complicates the analysis of the relaxation data.

[Figure]

Fig. S8: Hahn echo decay for bis-trityl **1** (4.1 nm) and bis-trityl **2** (5.3 nm) acquired with different pulse lengths. The oscillations are due to the dipolar coupling.

**S.4.2  Solvent deuteration: dOTP *vs* OTP**

We measured the Hahn echo decay of bis-trityl **1** in *ortho*-terphenyl (OTP) and its deuterated variant (dOTP). Surprisingly, we observed no difference between the two, see Fig. S9. Most likely, the protons in the trityl itself are responsible for the relaxation. Unfortunately, we do not (yet) have the fully deuterated variants of the used trityl moieties in order to test this hypothesis.

[Figure]

Fig. S9: Hahn echo decay of bis-trityl **1** acuqired with 100/200 ns pulses with OTP and dOTP as solvents. No significant difference can be observed between the two.

**S.4.3  SIFTER data for bis-trityls 1 and 2**

The SIFTER traces for bis-trityls **1** and **2** and corresponding distance distributions including a sensitvity analysis regarding the background and additional noise are shown in Fig. S10. The fitted background was included in the kernel function [1]. The distance distributions show the expected distances of around 4.1 and 5.3 nm. The oscillations in the case of bis-trityl **1** are quite damped compared to other model systems, and also compared to the dressed echo traces shown in the main text. It is possible that the compound shows strong or intermediate coupling effects for some orientations.

[Figure]

Fig. S10: SIFTER data for bis-trityls **1** and **2**. (**a**) Experimental SIFTER trace (black), symmetrized around the zero-time. Best fit including background in red. (**b**) Corresponding distance distributions obtained with the DeerLab API (`https://github.com/luisfabib/deerlab`). Black curve shows the median values for the probability densities. Red areas span from the 0.25 to 0.75 quantile.

**S.5 Additional data for figures in the main article**

Here we give detailed information on each figure. This includes information about raw data and processing scripts which are deposited online with a permanent DOI: `10.5281/zenodo.3703053`. Since our home-built spectrometer is run *via* MATLAB$^©$ (The MathWorks, Inc), even the raw data are stored in `.mat` files. Some scripts use functions from *EasySpin*[4]. Since *EasySpin*, written in MATLAB, is the most widely used simulation library in the EPR community, we believe that a majority of readers will be able to read and run the scripts we provide. Note that the `.mat` files can also be read into the free software R, with the help of the library 'R.matlab' (https://cran.r-project.org/web/packages/R.matlab/).

**Fig. 3 (a)**

Q-Band chirp echo Fourier transform EPR spectrum of mono-trityl **7**. 200 $\mu$M in dOTP. 6 $\mu$l of the solution in a 1.6 mm outer diameter tube. Temperature: 50 K. Pulse sequence: 200/100ns two-pulse chirp echo. Linear chirp pulses with total sweep width of 300 MHz, centered at the center of the resonator (34.8 GHz) and 20 ns rise time were used. The FM function was adjusted to compensate the resonator profile. Interpulse delay 2 $\mu$s (start-to-start of pulses). A symmetric 3.6 $\mu$s Chebyshev window with 100 dB relative sidelobe attenuation was applied to the digitally down-converted echo. A Fourier transform then yielded the spectrum. Magnetic field: 1241.4 mT (calibrated with DPPH). Shot repetition time 10 ms. 100 shots. 10 averages. $[(+)0, (-)180]$ phase cycle for the first pulse.
**Raw data and processing scripts:** Matlab processing and simulation: `spectrum_monomer.m`
raw file: `20191106_1239_chirp_echo_dummy.mat`

**Fig. 3 (b)**

Relaxation measurements for mono-trityl **7**. 200 $\mu$M in dOTP. 6 $\mu$l of the solution in a 1.6 mm outer diameter tube. Temperature: 50 K. Magnetic field: 1241.4 mT.

$T_m$ measurement: Two-pulse echo ($\pi/2 - \tau - \pi - \tau-$ echo) with 100/200 ns pulses. $[(+)0, (-)180]$ phase cycle for the first pulse. Inter-pulse delay varied from 200 ns to 16552 ns in steps of 32 ns (512 points in total). Note that the trace was to long and was thus cut for processing and plotting. Shot repetition time 30 ms. 10 shots per point. 1 average.

$T_{1\rho}$ measurement: $\pi/2 - \text{lock} - \tau - \pi - \tau-$echo. $(\pi/2)/\pi$ =4/8 ns. $\tau$=200 ns. Spin-lock Rabi frequency around 100 MHz. $[(+)0, (-)180]$ phase cycle for the $\pi/2$ pulse, $[(+)0, (+)180]$ phase cycle for the $\pi$ pulse. Fixed phase for the spin-lock, but always orthogonal to the first pulse. Spin-lock length varied from 0 ns to 100 ns in steps of 0.5 ns, and from 200 ns to 40 $\mu$s in steps of 100 ns (600 points in total). Shot repetition time 30 ms. 5 shots per point. 1 average.

$T_{2\rho}$ measurement: Dressed spin echo, see previous sections for detailed definitions. Dressed $(\pi/2)/\pi$ pulse lengths 21/42 ns, with an amplitude of 0.3 rad. Total spin-lock length 35 $\mu$s. Modulation frequency of 99 MHz. $\tau_0$ =2 $\mu$s, $\tau_1$ varied from 22 ns to 14390 ns in steps of 32 ns (450 points in total). Phase cycle and phase-phase cycle as introduced above. Shot repetition time 30 ms. 1 shot per point. 300 averages (overkill, was simply overnight). All other parameters as for the $T_{1\rho}$ measurement.
**Raw data and processing scripts:**

Matlab processing and simulation: `relaxation_monomers.m`
raw file $T_m$ measurement: `20200203_1538_2pecho_ESEEM.mat`
raw file $T_{1\rho}$ measurement: `20200203_1549_spin_locked_echo_t1rho.mat`
raw file $T_{2\rho}$ measurement: `20200203_1700_spin_locked_phase_echo_mod`

**Fig. 4 (a)**

Q-Band chirp echo Fourier transform EPR spectrum of bis-trityl **1**. 100 $\mu$M in dOTP. 6 $\mu$l of the solution in a 1.6 mm outer diameter tube. Temperature: 50 K. Pulse sequence: 200/100ns two-pulse chirp echo. Linear chirp pulses with total sweep width of 300 MHz, centered at the center of the resonator (34.8 GHz) and 20 ns rise time were used. The FM function was adjusted to compensate the resonator profile. Interpulse delay 950 ns (start-to-start of pulses). A symmetric 3.6 $\mu$s Chebyshev window with 100 dB relative sidelobe attenuation was applied to the digitally down-converted echo. A Fourier transform then yielded the spectrum. Magnetic field: 1241.4 mT (calibrated with DPPH). Shot repetition time 10 ms. 100 shots. 1 average. $[(+)0, (-)180]$ phase cycle for the first pulse.
**Raw data and processing scripts:** Matlab processing and simulation: `spectrum_4p1nm.m`
raw file: `20190729_1134_chirp_echo_dummy.mat`

**Fig. 4 (b)**

Relaxation measurements for bis-trityl **1**. 100 $\mu$M in dOTP. 6 $\mu$l of the solution in a 1.6 mm outer diameter tube. Temperature: 50 K. Magnetic field: 1241.4 mT.
$T_m$ measurement: Two-pulse echo ($\pi/2 - \tau - \pi - \tau -$ echo) with 100/200 ns pulses. $[(+)0, (-)180]$ phase cycle for the first pulse. Inter-pulse delay varied from 200 ns to 8376 ns in steps of 16 ns (512 points in total). Shot repetition time 30 ms. 10 shots per point. 1 average.
$T_{1\rho}$ measurement: $\pi/2 - \text{lock} - \tau - \pi - \tau -$echo. $(\pi/2)/\pi = 4/8$ ns. $\tau = 200$ ns. Spin-lock Rabi frequency around 100 MHz. $[(+)0, (-)180]$ phase cycle for the $\pi/2$ pulse, $[(+)0, (+)180]$ phase cycle for the $\pi$ pulse. Fixed phase for the spin-lock, but always orthogonal to the first pulse. Spin-lock length varied from 0 ns to 100 ns in steps of 0.5 ns, and from 200 ns to 40 $\mu$s in steps of 100 ns (600 points in total). Shot repetition time 30 ms. 5 shots per point. 1 average.
**Raw data and processing scripts:**
Matlab processing and simulation: `relaxation_4p1nm.m`
raw file $T_m$ measurement: `20191101_1415_2pecho_ESEEM.mat`
raw file $T_{1\rho}$ measurement: `20190819_1706_spin_locked_echo_t1rho.mat`

**Fig. 4 (c) and (d)**

Dressed echo decay/modulation for bis-trityl **1**. See previous sections for detailed definitions. 100 $\mu$M in dOTP. 6 $\mu$l of the solution in a 1.6 mm outer diameter tube. Temperature: 50 K. Magnetic field: 1241.4 mT. Dressed $(\pi/2)/\pi$ pulse lengths 21/42 ns, with an amplitude of 0.34 rad. Total spin-lock length 25 $\mu$s. Modulation frequency of 108 MHz. $\tau_0 = 1$ $\mu$s, $\tau_1$ varied from 22 ns to 8997 ns in steps of 25 ns (360 points in total). Phase cycle and phase-phase cycle as introduced above. Shot repetition

time 30 ms. 5 shot per point. 63 average. All other parameters as for the $T_{1\rho}$ measurement. Background correction with a stretched exponential and a Fourier transform gives the spectrum.

**Raw data and processing scripts:**

Matlab processing and simulation: `dressed_echo_modulation_4p1nm.m`

raw file: `20190820_1736_spin_locked_phase_echo_mod.mat`

**Fig. 5 (a)**

Q-Band chirp echo Fourier transform EPR spectrum of bis-trityl **2**. 100 $\mu$M in dOTP. 6 $\mu$l of the solution in a 1.6 mm outer diameter tube. Temperature: 50 K. Pulse sequence: 200/100ns two-pulse chirp echo. Linear chirp pulses with total sweep width of 300 MHz, centered at the center of the resonator (34.8 GHz) and 20 ns rise time were used. The FM function was adjusted to compensate the resonator profile. Interpulse delay 950 ns (start-to-start of pulses). A symmetric 3.6 $\mu$s Chebyshev window with 100 dB relative sidelobe attenuation was applied to the digitally down-converted echo. A Fourier transform then yielded the spectrum. Magnetic field: 1241.4 mT (calibrated with DPPH). Shot repetition time 10 ms. 100 shots. 1 average. $[(+)0,(-)180]$ phase cycle for the first pulse.

**Raw data and processing scripts:** Matlab processing and simulation: `spectrum_5p3.m`

raw file: `20191101_1155_chirp_echo_dummy.mat`

**Fig. 5 (b)**

Relaxation measurements for bis-trityl **2**. 100 $\mu$M in dOTP. 6 $\mu$l of the solution in a 1.6 mm outer diameter tube. Temperature: 50 K. Magnetic field: 1241.4 mT.

$T_m$ measurement: Two-pulse echo ($\pi/2 - \tau - \pi - \tau-$ echo) with 100/200 ns pulses. $[(+)0,(-)180]$ phase cycle for the first pulse. Inter-pulse delay varied from 200 ns to 8376 ns in steps of 16 ns (512 points in total). Shot repetition time 30 ms. 10 shots per point. 1 average.

$T_{1\rho}$ measurement: $\pi/2 - \mathrm{lock} - \tau - \pi - \tau-$echo. $(\pi/2)/\pi$ =4/8 ns. $\tau$=200 ns. Spin-lock Rabi frequency around 100 MHz. $[(+)0,(-)180]$ phase cycle for the $\pi/2$ pulse, $[(+)0,(+)180]$ phase cycle for the $\pi$ pulse. Fixed phase for the spin-lock, but always orthogonal to the first pulse. Spin-lock length varied from 0 ns to 100 ns in steps of 0.5 ns, and from 200 ns to 40 $\mu$s in steps of 100 ns (600 points in total). Shot repetition time 30 ms. 5 shots per point. 1 average.

**Raw data and processing scripts:**

Matlab processing and simulation: `relaxation_5p3nm.m`

raw file $T_m$ measurement: `20191101_1336_2pecho_ESEEM.mat`

raw file $T_{1\rho}$ measurement: `20191101_1058_spin_locked_echo_t1rho.mat`

**Fig. 5 (c) and (d)**

Dressed echo decay/modulation for bis-trityl **2**. See previous sections for detailed definitions. 100 $\mu$M in dOTP. 6 $\mu$l of the solution in a 1.6 mm outer diameter tube. Temperature: 50 K. Magnetic field: 1241.4 mT. Dressed $(\pi/2)/\pi$ pulse lengths 22/44 ns, with an amplitude of 0.3 rad. Total spin-lock length 35 $\mu$s. Modulation frequency of 108 MHz. $\tau_0$ =1 $\mu$s, $\tau_1$ varied from 22 ns to 14390 ns in steps

of 32 ns (450 points in total). Phase cycle and phase-phase cycle as introduced above. Shot repetition time 30 ms. 5 shot per point. 56 average. All other parameters as for the $T_{1\rho}$ measurement. Background correction with a stretched exponential and a Fourier transform gives the spectrum.

**Raw data and processing scripts:**

Matlab processing and simulation: `dressed_echo_modulation_5p3nm.m`
raw file: `20190821_1636_spin_locked_phase_echo_mod.mat`

**Fig. 6 (a) to (d): Simulations**

The simulation scripts are named *runsim_\*.m*. Where the asterisk describes the distance, offset distributions and Rabi frequencies. The actual simulation is done by the function *spinlock.m* . The corresponding plotting scripts are named *plot_\*.m*. (But you have to run the simulations first such that the results are saved).

**References**

1 Fábregas Ibáñez, L. and Jeschke, G.: Optimal background treatment in dipolar spectroscopy, Phys. Chem. Chem. Phys., 22, 1855–1868, https://doi.org/10.1039/C9CP06111H, 2020.

2 Henstra, A., Dirksen, P., Schmidt, J., and Wenckebach, W. T.: Nuclear spin orientation via electron spin locking (NOVEL), Journal of Magnetic Resonance (1969), 77, 389–393, https://doi.org/10.1016/0022-2364(88)90190-4, 1988.

3 Meyer, A., Jassoy, J. J., Spicher, S., Berndhäuser, A., and Schiemann, O.: Performance of PELDOR, RIDME, SIFTER, and DQC in measuring distances in trityl based bi- and triradicals: Exchange coupling, pseudosecular coupling and multi-spin effects, Physical Chemistry Chemical Physics, 20, 13 858–13 869, https://doi.org/10.1039/c8cp01276h, 2018.

4 Stoll, S. and Schweiger, A.: EasySpin, a comprehensive software package for spectral simulation and analysis in EPR, Journal of Magnetic Resonance, 178, 42–55, https://doi.org/10.1016/j.jmr.2005.08.013, 2006.

---

## Author Comment (AC3) · 14 Apr 2020

Review in Black
Response in Red
Manuscript parts in blue

The manuscript of Wili et al. describes a very new and exciting experiment to measure dipolar couplings between two trityl radicals at Q-band frequencies using spin-lock techniques and phase modulation two pulse echo of this 'dressed spin states' in the nutation frame. This work adds a new possibilities to prolong the observation time window for the measurement of dipolar couplings in EPR which could be useful to extend the distance range in the future. Despite the fact that the experimental problems seen (and described very clear and fair in the manuscript) do not allow routine application of this methods yet for long distances (where the dipolar coupling strength is much less than the inhomogenous linewidth) this very new approach is very interesting.

The experiments as well as the theoretical description and discussion is very good, the literature is cited appropriately and the existing problems with this very new approach – especially limitations with respect to Rabi oscillation frequency strength in comparison with the inhomogeneous linewidth – is fair and clear described. There are many interesting aspects in this work, as for example also the large difference between T1rho and T2rho, which will stimulate further work in this direction. I recommend publication of this very nice and innovative article.

We are pleased to see your positive review.

Some small remarks:
- Line 26 there should be an 'for' instead of 'or'

We corrected the mistake (actually Line 24).

- The exchange interaction is explicitly mentioned in the theoretical part; also the fact that in the interaction frame it might gain some additional importance because the Zeeman splitting and the linewidth offsets disappear. But then it is not mentioned any more. Of course the two model systems will not show such contributions, but maybe the authors have investigated potential effects of this theoretically? It would be nice to have a remark on this aspect in the discussion (or conclusion). As far as I see all trityls will be in the strong coupling regime, so the method could also work for shorter distances, where such interaction might play a role.

We did not perform any simulations, but we now added a sentence in the theory part. The paragraph on the exchange coupling now reads (starting from line 124 in the highlighted manuscript);

If, both, the dressed spin offsets as well as the spin states of the two spins are the same, exchange coupling has no influence on the evolution. This is analogous to the situation of magnetically equivalent nuclei in liquid state NMR. This different averaging of dipolar and exchange contributions might be exploited experimentally to distinguish the two contributions.

- It will be interesting to see what happens with the deuterated trityl radicals. As mentioned this will be something for a new publication and might shine some more light on the big differences between T1rho and T2rho. Also the T dependence of these rotating frame relaxation rates could be very interesting (also to further optimize the experiment)
We agree that rotating frame relaxation in EPR is not very well understood and that additional studies are desirable. Unfortunately, we did not systematically investigate the T-dependence of T1rho and T2rho. We did look at temperature dependence of the decay of the spin-locked echo, but this cannot easily be

translated to T1rho and T2rho. (See also the answers we will provide to the Review of Jack Freed: https://doi.org/10.5194/mr-2020-7-RC2).

While the relaxation in dOTP is interesting as well, regarding application work, the important matrices are water/glycerol and lipid bilayers. The room-temperature rotating-frame relaxation times of (immobilized) trityls would be interesting as well.

- The exponential 'stretch' factors for the fits of T1rho and T2rho should also be given.

We now report them in all figure captions where they are relevant.

- The modulation amplitude aPM was set to 0.3 for the experiments and it is also mentioned in the manuscript that the theoretical modelling brakes down if this factor becomes too large. Can this be more quantified?

We did not systematically quantify this in a way that we could specify an exact cut-off where the experiment fails. We looked at the phase-pulse nutation curve (Fig. S5) and took a value where the dressed pi-pulse length was not too long, but the additional oscillations were not too large. Some articles (*e.g.* Laucht *et al.* https://journals.aps.org/prb/abstract/10.1103/PhysRevB.94.161302) point out that the breakdown of the rotating wave approximation could be used to generate faster switching times, i.e. pi pulses. So far, we do not see a reliable way of doing this. Note also that we use a phase-phase cycle, which should cancel some contributions from imperfect dressed spin pulses.

---

## Author Comment (AC4) · 14 Apr 2020

Review in Black
Response in Red
Manuscript parts in blue

 The manuscript "Distance measurements between trityl radicals by pulse dressed electron paramagnetic resonance with phase modulation" by N. Wili, et al. describes a novel PDS method based on clever evolving dipolar coupling in the spin-locked state, thereby improving the distance range for trityl spin labels. The manuscript is generally correct and should be published with the consideration of the comments that we made.

This work describes an interesting development as so far it has not been demonstrated in ESR that spin-locked electron spins could be used to evolve selectively electron-electron dipolar coupling. The manuscript is well organized and clearly written. Also, extensive effort to synthesize and characterize rigid trityl biradicals and to simulate the evolution of coherence is shown. In particular, we like the implementation of the pulse sequence refocusing the nutation phase and providing the dipolar evolution sandwich in the locked state followed by the readout sequence, which can in principle exclude unwanted dipolar evolution. A simple two-pulse echo sequence used for readout appears sufficient at this stage, and by using SIFTER and DQC it would be possible to approach the somewhat shorter distance range. However, the dead-time cannot be excluded. Maybe the authors need to apply SIFTER or DQC sequences in the locked state.

We thank you for the generally positive comments. Regarding the dead time and other sequences in the locked state, you are correct. We added a comment in the manuscript, after the discussion of the pulse sequence (starting from line 241 in the highlighted manuscript):

If the dead time becomes too large for the relevant dipolar oscillations, one could, in principle, apply the known dead-time free single-frequency pulse sequences also as a phase-pulse sequence in the nutating frame.

It is encouraging that despite all the limitations imposed by the interaction strengths, the method does work and notably improves the dipolar evolution time for trityls. This work suggests a set of possible factors limiting T2(rho), and hopefully future work may be able to mitigate the effects of some of these factors. We doubt, however, that TWTA amplitude or phase noise contributes to shorter T2(rho). This is certainly not the case for amplitude noise, which within the locking bandwidth is estimated to be in the~1-10 mW range or maybe even less for a typical tube (less than -10 dBm/MHz noise spectral density). The phase noise of a TWTA (which is likely made by Applied System Engineering) is expected to be rather low. At least, the phase-noise test data for all amplifiers built over 20 years for ACERT supports this notion. Introducing phase noise, while possible, would be a complicated matter. AM/PM conversion in a saturated tube may be a possible way to test the effect on T1(rho) and T2(rho) to provide some insights on the instrumentation-imposed limits.

We thank you for the comments regarding the TWTA noise and sharing your experience with the noise figures. We based our comment on (Cohen et al. 2016: https://doi.org/10.1002/prop.201600071). (Which we now also clearly state in the manuscript). While we agree that the noise is rather small compared to the wanted driving field, it is not clear to us that one can safely conclude that it is irrelevant for T2rho. 10 mW is roughly a factor of 10000 smaller than the nominal 150 W nominal output power. This amounts to a factor of roughly 1% in terms of Rabi frequency, i.e. around 1 MHz. At least on first sight, it seems to be possible that this might contribute to T2rho.

Unfortunately, the work gives no clue regarding what to expect at a different temperature for T1(rho) and T2(rho). The experimental setup allows for easy temperature change and it is highly desirable to see the ratio of Tm/T2(rho) for at least one more temperature.

Under normal circumstances, we would simply measure it at another temperature and put the data in the SI. Unfortunately, with the current situation regarding COVID-19, we will be unable to provide these measurements in the coming months. We do not think that finalization of the paper should be held up that long.

The reason we did not measure at a different temperature so far is because we thought that a single additional temperature is not relevant. When comparing different temperatures, we also want to compare different matrices and measure rotating frame relaxation times at room temperature, i.e. for immobilized samples. Unfortunately, all of this must wait for the time being. It might also be too much for the scope of this manuscript. We do work on a more extended study on trityl relaxation.

At the beginning of the project we did measure decay of the spin-locked echo at different temperatures. However, as seen in Fig S3 of the submitted SI, this decay corresponds neither to T1rho nor to T2rho.

We hope that you still regard our proof-of-principle as relevant, although we measured only at a single temperature.

There is no comparison with the existing pulse sequences such as DQC or SIFTER,but we agree this may be unnecessary for this work. There is sufficient data in the literature for them, and PDS heavily relies on nitroxides anyway.

After emphasizing the power of major PDS methods in the introduction, the authors demonstrated that the sequence works at least for trityl labels, but they make no comment on whether it could be extended to any other known spin-label. The labels that are commonly used in biomedical research and are subjects of most of the key works cited are nitroxides, which demonstrate Tm's longer than the trityl's T2(rho) of this work. It is well known in this field that nitroxide labels quite often destabilize and precipitate proteins, the issue being even more critical with trityl labels, which are by no means mainstream. This is a significant limitation to the scope of this complex novel PDS method. Assuming that trityl labels were to have progressed to comparable use, there are other challenges that need to be addressed. The protein and lipid dynamics leading to Tm's in the low microsecond range as well as high local concentrations in the case of membrane proteins may contribute a set of problems in achieving T2ï Aš anywhere close to that observed in this work in dOTP glass. Note that the T2(rho)'s obtained are considerably shorter than for nitroxides in this glass. We have (unpublished) data that demonstrate $40\mu s$ evolution time in this glass using the DEER-5 method; 4-pulse DEER is also not very far from this mark. We also found very long Tm's for (partly) deuterated proteins (Georgieva et al., J. Biol. Chem., 2010). We think this work should be cited in the context of deuterated proteins.

We stressed in the article that the Rabi frequencies must be larger than the offsets. This is of course not possible for most labels other than trityls with current hardware. We now emphasize more that nitroxides are the most commonly used labels and that there are cases where nitroxide Tm is longer than the T2rho that we observe for trityls.

Changes:

Introduction (line 63 in highlighted manuscript): "Note that the sequence presented in this work relies on the narrow spectrum of the trityl radicals. We do not expect it to work with the much more commonly used nitroxide radicals."

Conclusion(line 415): "Note that in dOTP, the Tm values of the slow relaxing component of nitroxides (the relaxation of nitroxides in dOTP can be described by a sum of two stretched exponentials) can still be bigger than the T2rho times measured here for trityl radicals (Soetbeer2018)."

The mentioned paper by Georgieva et al. is now cited. We were unaware that it preceded the work by Ward et al. in 2010, and it is indeed relevant when discussing protein deuteration and PDS. We apologize for the oversight.

Regarding all the other issues: We agree that there are most likely other hurdles. Lipid dynamics might indeed contribute to shorter T2rho values – or it may not (some dynamics can indeed be decoupled in rotating frame experiments in NMR). Only experiment will tell. We publish our findings early to encourage other interested researchers to help answer these questions.

We clearly state in the conclusions that the sequence is not yet ready for application work. We believe that this statement is sufficiently strong in pointing out current limitations.

The special technical requirements of this experiment to provide long intense locking pulses need to be described in greater detail. Such pulses are not normally used in pulse ESR. A 150 W TWTA was employed to achieve~100 MHz Rabi frequency (36 G B1) and this power could last for about 40μs periods limited by the amplifier. In the first place it is a lot of power that can cause heating, arcing, and a damage to the receiver, thus limiting the repetition frequency. How was the receiver protected?

No special measures beyond the ones reported in (Doll & Jeschke, 2017) (now cited) were taken to protect the receiver – we simply made sure we had a spare switch and low-noise amplifier ready in case we destroy them. However, this has not happened so far (We started doing the experiments in August 2019). We also let it run overnight. We found no reduction in performance so far.

However, your point is now emphasized in the manuscript, in order to alert researchers who want to use this sequence or similar sequences that their hardware might be at risk.

We added the following paragraph in the Materials and Methods section (line 255):

Note that the long spin-lock pulses with full power can be dangerous for the receiver, since much of the power is reflected by the overcoupled resonator. We did not take any special measures beyond the receiver protection switch (Doll & Jeschke 2017). However, we are planning to install an additional limiter or a slow switch that could take more power. Since the spin-lock pulses are rather long, a slow switch could be used for most of the time, while the fast switch could be used for the transient times of the pulses to still provide the small dead time.

The origins of the baseline and of modulation depth need to be discussed.

We added the following lines in the results section:

Note that not only intermolecular dipolar couplings from remote spins contribute to the background. Transverse relaxation of dressed spins with time constant T2rho also contributes because we do not perform a constant time experiment. Regarding modulation depth we would have expected it to be unity, which is clearly not seen in our experiments. We suspect that imperfections in the dressed spin pi-pulse lead to an unmodulated background, which cannot be

removed by phase-cycling. The phenomenon is similar to reduced instantaneous diffusion for a Hahn Echo if the flip angle of the pi-pulse is reduced.

Figure S8 – legends need be corrected.

Thank you for pointing this out. We corrected the legend that was mixed up.

---

## Author Comment (AC5) · 22 Apr 2020

We would like to clarify our comments regarding the temperature dependence:

This is an important point, and as we are mentioning in the outlook, we are planning a more comprehensive study of trityl relaxation with different degrees of deuteration of the trityl itself and the solvent. We are also planning to compare Tm with T2rho in different environments and then also at different temperatures.
* * *

---

## Author Response (AR2)

Review in Black Response in Red Manuscript parts in blue

The manuscript of Wili et al. describes a very new and exciting experiment to measure dipolar couplings between two trityl radicals at Q-band frequencies using spin-lock techniques and phase modulation two pulse echo of this 'dressed spin states' in the nutation frame. This work adds a new possibilities to prolong the observation time window for the measurement of dipolar couplings in EPR which could be useful to extend the distance range in the future. Despite the fact that the experimental problems seen (and described very clear and fair in the manuscript) do not allow routine application of this methods yet for long distances (where the dipolar coupling strength is much less than the inhomogenous linewidth) this very new approach is very interesting.

The experiments as well as the theoretical description and discussion is very good, the literature is cited appropriately and the existing problems with this very new approach – especially limitations with respect to Rabi oscillation frequency strength in comparison with the inhomogeneous linewidth – is fair and clear described. There are many interesting aspects in this work, as for example also the large difference between T1rho and T2rho, which will stimulate further work in this direction. I recommend publication of this very nice and innovative article.

**We are pleased to see your positive review.**

Some small remarks: - Line 26 there should be an 'for' instead of 'or'

**We corrected the mistake (actually Line 24).**

- The exchange interaction is explicitly mentioned in the theoretical part; also the fact that in the interaction frame it might gain some additional importance because the Zeeman splitting and the linewidth offsets disappear. But then it is not mentioned any more. Of course the two model systems will not show such contributions, but maybe the authors have investigated potential effects of this theoretically? It would be nice to have a remark on this aspect in the discussion (or conclusion). As far as I see all trityls will be in the strong coupling regime, so the method could also work for shorter distances, where such interaction might play a role.

We did not perform any simulations, but we now added a sentence in the theory part. The paragraph on the exchange coupling now reads (starting from line 124 in the highlighted manuscript);

If, both, the dressed spin offsets as well as the spin states of the two spins are the same, exchange coupling has no influence on the evolution. This is analogous to the situation of magnetically equivalent nuclei in liquid state NMR. This different averaging of dipolar and exchange contributions might be exploited experimentally to distinguish the two contributions.

- It will be interesting to see what happens with the deuterated trityl radicals. As mentioned this will be something for a new publication and might shine some more light on the big differences between T1rho and T2rho. Also the T dependence of these rotating frame relaxation rates could be very interesting (also to further optimize the experiment)

We agree that rotating frame relaxation in EPR is not very well understood and that additional studies are desirable. Unfortunately, we did not systematically investigate the T-dependence of T1rho and T2rho. We did look at temperature dependence of the decay of the spin-locked echo, but this cannot easily be

translated to T1rho and T2rho. (See also the answers we will provide to the Review of Jack Freed: https://doi.org/10.5194/mr-2020-7-RC2).

While the relaxation in dOTP is interesting as well, regarding application work, the important matrices are water/glycerol and lipid bilayers. The room-temperature rotating-frame relaxation times of (immobilized) trityls would be interesting as well.

- The exponential 'stretch' factors for the fits of T1rho and T2rho should also be given.

We now report them in all figure captions where they are relevant.

- The modulation amplitude aPM was set to 0.3 for the experiments and it is also mentioned in the manuscript that the theoretical modelling brakes down if this factor becomes too large. Can this be more quantified?

We did not systematically quantify this in a way that we could specify an exact cut-off where the experiment fails. We looked at the phase-pulse nutation curve (Fig. S5) and took a value where the dressed pi-pulse length was not too long, but the additional oscillations were not too large. Some articles (*e.g.* Laucht *et al.* https://journals.aps.org/prb/abstract/10.1103/PhysRevB.94.161302) point out that the breakdown of the rotating wave approximation could be used to generate faster switching times, i.e. pi pulses. So far, we do not see a reliable way of doing this. Note also that we use a phase-phase cycle, which should cancel some contributions from imperfect dressed spin pulses.

Review in Black Response in Red Manuscript parts in blue

The manuscript "Distance measurements between trityl radicals by pulse dressed electron paramagnetic resonance with phase modulation" by N. Wili, et al. describes a novel PDS method based on clever evolving dipolar coupling in the spin-locked state, thereby improving the distance range for trityl spin labels. The manuscript is generally correct and should be published with the consideration of the comments that we made.

This work describes an interesting development as so far it has not been demonstrated in ESR that spin-locked electron spins could be used to evolve selectively electron-electron dipolar coupling. The manuscript is well organized and clearly written. Also, extensive effort to synthesize and characterize rigid trityl biradicals and to simulate the evolution of coherence is shown. In particular, we like the implementation of the pulse sequence refocusing the nutation phase and providing the dipolar evolution sandwich in the locked state followed by the readout sequence, which can in principle exclude unwanted dipolar evolution. A simple two-pulse echo sequence used for readout appears sufficient at this stage, and by using SIFTER and DQC it would be possible to approach the somewhat shorter distance range. However, the dead-time cannot be excluded. Maybe the authors need to apply SIFTER or DQC sequences in the locked state.

We thank you for the generally positive comments. Regarding the dead time and other sequences in the locked state, your correct. We added a comment in the manuscript, after the discussion of the pulse sequence (starting from line 241 in the highlighted manuscript):

If the dead time becomes too large for the relevant dipolar oscillations, one could in principle apply the known single-frequency pulse sequences also as a phase-pulse sequence in the nutating frame.

It is encouraging that despite all the limitations imposed by the interaction strengths, the method does work and notably improves the dipolar evolution time for trityls. This work suggests a set of possible factors limiting T2(rho), and hopefully future work may be able to mitigate the effects of some of these factors. We doubt, however, that TWTA amplitude or phase noise contributes to shorter T2(rho). This is certainly not the case for amplitude noise, which within the locking bandwidth is estimated to be in the~1-10 mW range or maybe even less for a typical tube (less than -10 dBm/MHz noise spectral density). The phase noise of a TWTA (which is likely made by Applied System Engineering) is expected to be rather low. At least, the phase-noise test data for all amplifiers built over 20 years for ACERT supports this notion. Introducing phase noise, while possible, would be a complicated matter. AM/PM conversion in a saturated tube may be a possible way to test the effect on T1(rho) and T2(rho) to provide some insights on the instrumentation-imposed limits.

We thank you for the comments regarding the TWTA noise and sharing your experience with the noise figures. We based our comment on (Cohen et al. 2016:

https://doi.org/10.1002/prop.201600071). (Which we now also clearly state in the manuscript). While we agree that the noise is rather small compared to the wanted driving field, we cannot follow you how you immediately conclude that it is irrelevant for T2rho. 10mW is roughly a factor of 10000 smaller than the nominal 150W nominal output power. This amounts to a factor of roughly 1% in terms of Rabi frequency, i.e. around 1 MHz. At least on first sight, it seems to be possible that this might contribute to T2rho.

Unfortunately, the work gives no clue regarding what to expect at a different temperature for T1(rho) and T2(rho). The experimental setup allows for easy temperature change and it is highly desirable to see the ratio of Tm/T2(rho) for at least one more temperature.

Under normal circumstances, we would simply measure it at another temperature and put the data in the SI. Unfortunately, with the current situations regarding COVID-19, we will be unable to provide these measurements in the coming months.

The reason we did not measure at a different temperature so far is because we thought that a single additional temperature is not relevant. If we compare different temperatures, we also wanted to compare different matrices, and we would also like to measure rotating frame relaxation times at room temperature, i.e. for immobilized samples. Unfortunately, all of this must wait for the time being.

What we did measure at the beginning of the project, was the decay of the spinlocked-echo at different temperatures. However, as seen in Fig S3 of the submitted SI, this decay neither corresponds to T1rho nor to T2rho.

We hope that you still regard this proof-of-principle as relevant, although it was only measured at one temperature.

We adjusted the following sentence in the outlook to make it clear that the temperature dependence is an important point:

Preliminary results with the OX063 trityl and its partially deuterated analogue OX71 in different solvent compositions (not shown) revealed that even bare-spin relaxation at low temperatures and low concentrations is complicated to understand, let alone dressed-spin relaxation with characteristic times T2rho and T1rho. We are planning to investigate this in more detail and to compare the different relaxation times also at different temperatures.

There is no comparison with the existing pulse sequences such as DQC or SIFTER, but we agree this may be unnecessary for this work. There is sufficient data in the literature for them, and PDS heavily relies on nitroxides anyway.

After emphasizing the power of major PDS methods in the introduction, the authors demonstrated that the sequence works at least for trityl labels, but they make no comment on whether it could be extended to any other known spin-label. The labels that are commonly used in biomedical research and are subjects of most of the key works cited are nitroxides, which demonstrate Tm's longer than the trityl's T2(rho) of this work. It is well known in this field that nitroxide labels guite often destabilize and precipitate proteins, the issue being even more critical with trityl labels, which are by no means mainstream. This is a significant limitation to the scope of this complex novel PDS method. Assuming that trityl labels were to have progressed to comparable use, there are other challenges that need to be addressed. The protein and lipid dynamics leading to Tm's in the low microsecond range as well as high local concentrations in the case of membrane proteins may contribute a set of problems in achieving T2ï Aš anywhere close to that observed in this work in dOTP glass. Note that the T2(rho)'s obtained are considerably shorter than for nitroxides in this glass. We have (unpublished) data that demonstrate 40µs evolution time in this glass using the DEER-5 method; 4-pulse DEER is also not very far from this mark. We also found very long Tm's for (partly) deuterated proteins (Georgieva et al., J. Biol. Chem., 2010). We think this work should be cited in the context of deuterated proteins.

We stressed in the article that the Rabi frequencies must be larger than the offsets. This is of course not possible for most other labels than trityls at the moment. We now emphasize more that nitroxides are the most commonly used labels and that there are cases where nitroxide Tm are longer than the T2rho that we observe for trityls.

**Changes:**

Introduction (line 63 in highlighted manuscript): "Note that the sequence presented in this work relies on the narrow spectrum of the trityl radicals. We do not expect it to work with the much more commonly used nitroxide radicals."

Conclusion(line 415): "Note that in dOTP, the Tm values of the slow relaxing component of nitroxides (the relaxation of nitroxides in dOTP can be described by a sum of two stretched exponentials) can still be bigger than the T2rho times measured here for trityl radicals (Soetbeer2018)."

The mentioned paper by Georgieva et al. is now cited. I was unaware that this was done earlier than the work by Ward et al. in 2010, and it is indeed relevant when discussing protein deuteration and PDS.

Regarding all the other issues: We agree that there are most likely other hurdles. Lipid dynamics might indeed contribute to shorter T2rho values. But maybe they do not (some dynamics can indeed be decoupled in rotating frame experiments in NMR). Only experiments will tell. We publish the findings in this articles such that other interested researchers can help to answer these questions if they want to.

**We clearly state in the conclusions that the sequence is not ready for application work. We believe this notion is strong enough regarding limitations.**

The special technical requirements of this experiment to provide long intense locking pulses need to be described in greater detail. Such pulses are not normally used in pulse ESR. A 150 W TWTA was employed to achieve~100 MHz Rabi frequency (36 G B1) and this power could last for about  $40\mu$ s periods limited by the amplifier. In the first place it is a lot of power that can cause heating, arcing, and a damage to the receiver, thus limiting the repetition frequency. How was the receiver protected?

We have to confess that no special measures were taken to protect the receiver – we simply made sure we had a spare switch and low-noise amplifier ready in case we destroy them. However, this has not happened so far (We started doing the experiments in August 2019). We also let it run overnight. We found no reduction in performance so far.

However, your point should be emphasized in the manuscript, such that researchers who want to use this or similar sequences are aware that their hardware might be at risk.

We added the following paragraph in the Materials and Methods section (line 255):

Note that the long spin-lock pulses with full power are a danger for the receiver of the spectrometer. We did not take any special measures beyond the usual protection switch of our spectrometer. However, we are planning to install an additional limiter or a slow switch that could take more power. Since the spin-lock pulses are rather long, a slow switch could be used for most of the time, while the fast switch could be used for the transient times of the pulses to still provide the small dead time.

The origins of the baseline and of modulation depth need to be discussed.

**We added the following lines in the results section:**

Note that the background here has not only contributions from intermolecular dipolar couplings (remote spins), but also a relaxation component of T2rho because we do not perform a constant time experiment. Theoretically, we would also expect a modulation depth of unity, which is clearly not the case experimentally. We suspect that imperfections in the dressed spin pi-pulse lead to an unmodulated background, which cannot be removed by phase-cycling. The phenomenon is similar to reduced instantaneous diffusion for a Hahn Echo if the flip angle of the pi-pulse is reduced.

Figure S8 – legends need be corrected.

Thank you for pointing this out. We corrected the legend that was mixed up.

[revised manuscript text omitted]
}}'_{mw} + \hat{\mathcal{H}}'_{offset} + \hat{\mathcal{H}}'_{e-e} + \hat{\mathcal{H}}'_{e-n} + \hat{\mathcal{H}}'_{nuc}$$
 (1)

The first term is the microwave Hamiltonian, which is given in the electron-spin rotating frame by

$$\hat{\mathcal{H}}'_{\rm mw} = \omega_1 \left( \hat{S}_{1,x} + \hat{S}_{2,x} \right) \text{ with } \omega_1 = -\gamma_e B_1 \quad .$$
(2)

The Rabi or nutation frequency is denoted by  $\omega_1$ , which depends on the microwave amplitude  $B_1$  and the gyromagnetic ratio of the electron,  $\gamma_e$ . We assume a constant microwave phase and neglect the influence of the microwaves on the nuclear spins. In the following, we will apply an interaction frame transformation (IAT) with  $\hat{\mathcal{H}}'_{mw}$  to all other terms and use first-order average Hamiltonian theory to gain physical insight. The new frame is referred to as the nutating frame. The nutating frame Hamiltonian is based on spin operators for dressed electron spins and bare nuclear spins. For mathematical details please consult the SI.

If we choose the nutating frame frequency  $\omega_{PM}$  equal to the Rabi frequency,  $\omega_{PM} = \omega_1$ , the irradiation term is completely absorbed into the frame. In a real experiment with an ensemble of spins,  $\omega_1$  will be distributed due to microwave inhomogeneities,

90 thus we will always have a remaining contribution of

$$\hat{\mathcal{H}}_{\rm mw}^{\prime\prime} = \Omega_{\rm d} \left( \hat{S}_{1,x} + \hat{S}_{2,x} \right) \text{ with } \Omega_{\rm d} = (\omega_1 - \omega_{\rm PM}) \quad . \tag{3}$$

The dressed spin offset  $\Omega_d$  will be distributed over the sample, but as a molecule is by orders of magnitude smaller than the microwave wavelength,  $\Omega_d$  will be the same for all electron spins within one molecule.

As usual, the influence of a small g-anisotropy and of an inhomogeneous static magnetic field  $B_0$  is captured in offset terms 95 in the rotating frame

$$\hat{\mathcal{H}}_{\text{offset}}' = \Omega_{S,1} \hat{S}_{1z} + \Omega_{S,2} \hat{S}_{2z} \quad . \tag{4}$$

We neglect any tilt of the electron spin quantization axis due to strong *g*-anisotropy, which is a good approximation for trityl and other organic radicals. The first-order average Hamiltonian after an IAT with  $\hat{\mathcal{H}}'_{mw}$  vanishes, *i.e.*

$$\hat{\mathcal{H}}_{\text{offset}}^{\prime\prime} = 0 \quad . \tag{5}$$

100 In pulse EPR, the spectral width is often much larger than the Rabi frequency. In this case, the first order approximation will be poor. It is, however, not a poor approximation for trityl radicals with our setup. For simplicity, we will mostly neglect the effect of resonance offsets  $\Omega_{S,1}$  and  $\Omega_{S,2}$ .

The most important term in the context of distance measurements is the electron-electron coupling Hamiltonian, which contains dipolar and exchange (J) contributions

105
$$\hat{\mathcal{H}}_{e-e}' = \hat{\mathcal{H}}_{e-e,dip}' + \hat{\mathcal{H}}_{e-e,J}'$$
$$\hat{\mathcal{H}}_{e-e,dip}' = \omega_{dd} \left( \hat{S}_{1z} \hat{S}_{2z} - \frac{1}{2} \left( \hat{S}_{1x} \hat{S}_{2x} + \hat{S}_{1y} \hat{S}_{2y} \right) \right)$$
$$\omega_{dd} = \frac{\mu_0}{4\pi} \frac{\mu_B^2 g_1 g_2}{\hbar} \frac{1}{r_{12}^3} \left( 1 - 3\cos^2 \theta \right)$$
$$\hat{\mathcal{H}}_{e-e,J}' = J \left( \hat{S}_1 \cdot \hat{S}_2 \right)$$
(6)

where  $\mu_0$  is the vacuum permeability,  $\mu_B$  the Bohr magneton,  $g_1$  and  $g_2$  are the *g*-factors of the two electron spins, and  $\theta$  is 110 the angle between the external magnetic field and the interspin vector with length  $r_{12}$ . The exchange contribution is often, but not always negligible in pulse EPR based distance measurements. The prefactor of the dipolar coupling contains the distance information and is given by

$$d = \frac{1}{2\pi} \frac{\mu_0}{4\pi} \frac{\mu_B^2 g^2}{\hbar} \frac{1}{r^3} \quad .$$
(7)

This amounts to 52.04 MHz for r = 1 nm. After transformation to the nutating frame, we obtain

115
$$\hat{\mathcal{H}}_{e-e,dip}'' = -\frac{1}{2} \cdot \omega_{dd} \left( \hat{S}_{1x} \hat{S}_{2x} - \frac{1}{2} \left( \hat{S}_{1z} \hat{S}_{2z} + \hat{S}_{1y} \hat{S}_{2y} \right) \right) \\ \hat{\mathcal{H}}_{e-e,J}'' = \hat{\mathcal{H}}_{e-e,J}' = J \left( \hat{S}_{1} \cdot \hat{S}_{2} \right) \quad .$$
(8)

[revised manuscript text omitted]
) K2CO3, MeOH, CH2Cl2, rt, 14.5 h, 96%; (c) PdCl2(PPh3)2, CuI, piperidine, THF, air, rt, 16 h, 36%; (d) SOCl2, CHCl3, 50 °C, 90 min, not isolated; (e) *i*Pr2NEt, CHCl3, rt, 17 h, 40%. For n = 2: (a) PdCl2(PPh3)2, CuI, piperidine, THF, rt, 46 h, 86%; (b) K2CO3, MeOH, CH2Cl2, rt, 14.5 h, 96%; (c) PdCl2(PPh3)2, CuI, piperidine, THF, rt, 46 h, 86%; (b) K2CO3, MeOH, CH2Cl2, rt, 14.5 h, 96%; (c) PdCl2(PPh3)2, CuI, piperidine, THF, air, rt, 15.5 h, 65%; (d) SOCl2, CHCl3, 50 °C, 90 min, not isolated; (e) *i*Pr2NEt, CHCl3, rt, 19 h, 64%. For further details see the SI part B. THF = tetrahydrofuran, TIPS = triisopropylsilyl, TMS = trimethylsilyl, rt = room temperature.

**4.3 Bis-trityl 1, $r \approx 4.1$ nm**

The results for bis-trityl **1** are shown in Fig. 4. The chirp echo FT-EPR spectrum is shown in panel (a). The spectrum consists of a slightly asymmetric line with an FWHM of 16 MHz. The theoretical excitation profile of a 4 ns and an 8 ns microwave pulse are overlaid, showing that the whole spectrum can be excited almost uniformly with rectangular pulses.

Figure 3. Measurements on mono-trityl 7. (a) EPR spectrum. The excitation profiles of the rectangular pulses used are indicated. They are sufficiently strong to excite the whole EPR line. The red dashed lines indicate a simulation based on the *g*-values given in (Hintz et al., 2019) and an Gaussian broadening of 8 MHz FWHM. (b) Corresponding echo decay curves. Experimental points in circles (not all points shown for clarity), and best fit in solid lines. The fitted values are  $T_m = 2.9 \,\mu\text{s}$  ( $\xi = 5.9$ ),  $T_{2\rho} = 13.1 \,\mu\text{s}$  ( $\xi = 4.6$ ) and  $T_{1\rho} = 930 \,\mu\text{s}$  ( $\xi = 2.4$ ).